# REVISEVAL: IMPROVING LLM-AS-A-JUDGE VIA RESPONSE-ADAPTED REFERENCES

**Qiyuan Zhang**[1]**, Yufei Wang**[2]**, Tiezheng Yu**[2]**, Yuxin Jiang**[3]**, Chuhan Wu**[2]**, Liangyou Li**[2]**,**
**Yasheng Wang**[2]**, Xin Jiang**[2]**, Lifeng Shang**[2]**, Ruiming Tang**[2]**, Fuyuan Lyu**[4]**, Chen Ma**[1]
[1]City University of Hong Kong, [2]Huawei Noah's Ark Lab,
[3]The Hong Kong University of Science and Technology (Guangzhou),
[4]McGill University & MILA
`qzhang732-c@my.cityu.edu.hk, wang.yufei1@huawei.com`

## ABSTRACT

With significant efforts in recent studies, LLM-as-a-Judge has become a cost-effective alternative to human evaluation for assessing text generation quality in a wide range of tasks. However, there still remains a reliability gap between LLM-as-a-Judge and human evaluation. One important reason is the lack of guided oracles in the evaluation process. Motivated by the role of *reference* pervasively used in classic text evaluation, we introduce REVISEVAL, a novel text generation evaluation paradigm via the response-adapted references. REVISEVAL is driven by the key observation that an ideal reference should maintain the necessary relevance to the response to be evaluated. Specifically, REVISEVAL leverages the text revision capabilities of large language models (LLMs) to adaptively revise the response, then treat the revised text as the reference (*response-adapted reference*) for the subsequent evaluation. Extensive experiments demonstrate that REVISEVAL outperforms traditional reference-free and reference-based evaluation paradigms that use LLM-as-a-Judge across NLG tasks and open-ended instruction-following tasks. More importantly, our response-adapted references can further boost the classical text metrics, *e.g.*, BLEU and BERTScore, compared to traditional references and even rival the LLM-as-a-Judge. A detailed analysis is also conducted to confirm REVISEVAL's effectiveness in bias reduction, the impact of inference cost, and reference relevance.

## 1 INTRODUCTION

As the large language model (LLM) already exhibits strong alignment with humans (Gilardi et al., 2023; OpenAI et al., 2024), LLM-as-a-Judge (Chang et al., 2024; Li et al., 2024b; Gao et al., 2024b), *aka*. LLM-evaluator, has emerged as a viable alternative to human evaluation in assessing text generation quality. Given the task instruction and the corresponding model-generated responses, LLMs are prompted to predict preferences or scores for these responses. Despite considerable efforts have been made, such as chain-of-thought (Zheng et al., 2023), specialized rubrics (Liu et al., 2023), and extensive evaluation-specific training datasets (Li et al., 2024a; Wang et al., 2024c;b), human evaluation remains the gold standard in text quality assessment (Zeng et al., 2024) and LLM-as-a-Judge struggles with particular biases (Huang et al., 2024) and being vulnerable to the misleading context (Dubois et al., 2024; Chen et al., 2024). One important reason is the lack of an oracle to direct the evaluation process. Fortunately, classical text evaluation metrics, like BLEU (Papineni et al., 2002) and ROUGE (Lin, 2004), offer valuable prompts in mitigating such a gap: given an appropriate reference, *i.e.*, the ground-truth answer to the task, calculating the similarity between the references and the model-generated responses can achieve a satisfactory correlation with human evaluations. Furthermore, several studies highlight the reference could prevent being overly sensitive to semantic deficiency (Sheng et al., 2024) and overcoming bias (Deutsch et al., 2022) in certain cases.

However, straightforward leveraging references in LLM-as-a-Judge can also be challenging. In addition to the availability of high-quality references (Rei et al., 2021), previous works (Mehri & Eskenazi, 2020; Gómez-Rodríguez & Williams, 2023; Guan & Huang, 2020) find that pre-existing references introduce noise across various text evaluation tasks due to the *one-to-many* problem,

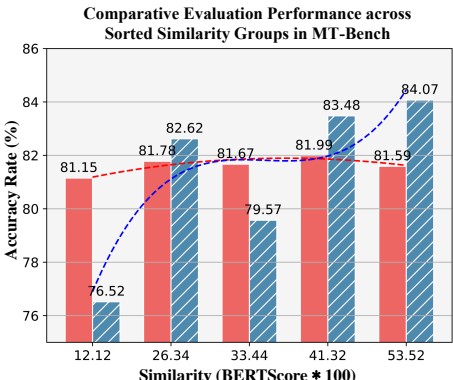

Figure 1: Performance comparison of reference-free and reference-based evaluation paradigms across different similarity groups in MT-Bench, using GPT-4-as-a-Judge. In the reference-based evaluation, the GPT-4 direct response is used as the reference, and the evaluated response with a higher BERTScore with the reference is regarded as the preferred one. As the similarity between the reference and the response increases, the human agreement accuracy of the reference-based evaluation significantly improves, while the reference-free evaluation maintains relatively consistent performance across all similarity levels.

where for a given task input, there exist many diverse yet valid responses. In this case, particular pre-existing references could negatively penalize many appropriate but dissimilar responses in the evaluation process (Ji et al., 2022). Thus, we hypothesize that *an effective reference should be closely relevant to the response to be evaluated*. We further verify this on MT-Bench (Zheng et al., 2023), an open-ended instruction-following dataset. As shown in Figure 1, we use GPT-4 direct responses to the instructions as the references and quantify relevance by the similarity between the references and the responses using BERTScore (Zhang et al., 2020). We find that higher relevance simulates greater utility from the reference, resulting in more effective evaluations than a reference-free evaluator.

Motivated by the above findings, we deem that an effective reference should maintain **high quality** while ensuring **relevance** to the response, which led us to consider that revising the response adaptively could be a good candidate for this reference (Guo et al., 2024). Therefore, we propose **a novel evaluation paradigm REVISE-AND-EVALUATION (REVISEVAL)**. Specifically, given the (instruction, response) pair, REVISEVAL first revise the response using the instruction and evaluation rubric, resulting in the *response-adapted reference*. REVISEVAL further leverages this generated *response-adapted reference* to guide final evaluation (*e.g.*, scoring or pairwise comparison). By revising the original response, we ensure that the generated reference is both high-quality and closely relevant to the original content. The comparison between the original and revised responses offers valuable insights for evaluation. Orthogonal to previous work that only focuses on the discrimination of LLMs, REVISEVAL stands out by fully utilizing the generative potential by revision.

We conduct comprehensive experiments to validate the effectiveness of our proposed REVISEVAL. Using both proprietary and open-source LLMs, REVISEVAL consistently achieves better performance compared to reference-free and reference-based evaluation paradigms in both NLG tasks and open-ended instruction-following tasks. Moreover, we seek to verify the effectiveness of the response-adapted references in the classic metrics, *e.g.*, BERT and BERTScore, showing that each metric exceeds itself by up to 3%-10% accuracy compared to using direct response as references. We then combine LLM-as-a-reviser with multiple classic metrics and find that it outperforms (reference-free) LLM-as-a-Judge by over 1.5% on average when using weak LLMs and is comparable when using GPT-4. Finally, we analyze how our paradigm achieves overall superiority. In reducing verbosity and positional bias, our approach offers clear advantages in adversarially designed LLMBar and swapping position testing. Merely increasing inference cost of reference-free evaluation still lags behind REVISEVAL, demonstrating our method's efficiency does not rely on naively accumulating cost. Meanwhile, we validate the relationship between reference relevance and efficiency using response-adapted references.

## 2 RELATED WORK

### 2.1 EVALUATION OF LARGE LANGUAGE MODELS

Instruction-tuned LLMs (Ouyang et al., 2022; Dubey et al., 2024; Team et al., 2023) have revolutionized the field of NLP due to their ability to handle a wide range of language-related tasks. Unlike traditional NLP tasks, such as Machine Translation and Summarization, which can be eval-

uated using N-gram-based metrics like BLEU (Papineni et al., 2002), ROUGE (Lin, 2004), and METEOR (Banerjee & Lavie, 2005) by comparing responses with reference texts, LLMs excel at open-ended text generation tasks (*e.g.*, story generation and instruction-following generation), where no single reference response exists. Consequently, several studies embrace the potential of LLM-as-a-Judge and shift toward reference-free metrics, advocating the abandonment of conventional reference-based evaluation methods (Sheng et al., 2024; Chen et al., 2023). In this paper, we revisit the significance of reference in LLM evaluation. Furthermore, to address the challenge of the absence of a single standard answer in certain evaluation tasks, we propose leveraging LLMs to generate response-adapted references, thereby improving the performance of both traditional metrics and LLM-as-a-Judge.

## 2.2 LLM-AS-A-JUDGE

Recent progress in NLP has introduced model-based evaluation metrics like BERTScore (Zhang et al., 2020) and BARTScore (Yuan et al., 2021). However, these methods also depend on the availability of human-annotated references, which can be expensive, time-consuming, and labor-intensive (Zheng et al., 2023). With the emergence of large language models (LLMs), several studies (Zheng et al., 2023; Dubois et al., 2024) have harnessed their robust evaluation capabilities for assessing natural language generation (NLG), particularly by employing proprietary models like GPT-4 (OpenAI et al., 2024). To avoid information leakage caused by external API calls, some efforts advocate finetuning LLMs with evaluation data to obtain evaluator models (Vu et al., 2024; Li et al., 2024a; Wang et al., 2024c; Kim et al., 2024b). A wide variety of techniques are used to enhance the performance of LLM-as-a-Judge, such as Chain-of-Thoughts (CoT; Wei et al. (2022)) to first generate concise reasoning and then the final decision, adding pre-defined rules (Zeng et al., 2024) in prompts to list some general rules for LLM-as-a-Judge to follow explicitly, and swapping the two responses to avoid positional bias (Wang et al., 2024a). Zheng et al. (2023) found that, even with the use of a CoT prompt, LLM-as-a-Judge can still be misled by the surrounding context, particularly by erroneous response text. Therefore, they propose a reference-guided method where the LLM-as-a-Judge's response is first generated independently based on the given instruction and then presented as a reference answer within the evaluation prompt. To the best of our knowledge, we are the first to generate response-adapted references from both the instruction and the response to be evaluated.

## 3 METHODOLOGY

In this section, we introduce our novel evaluation paradigm, REVISEVAL, which enhances the evaluation by generating response-adapted references. Illustrated in Figure 2, REVISEVAL consists of two components, **response-adapted reference generation** and **reference-based evaluation**, which we will discuss in Sec. 3.1 and 3.2, respectively.

Supposing $y$ is the response generated by a model for a given task instruction $x$, REVISEVAL assesses the quality of $y$ on a specific rubric $a$. Firstly, in the **generation** phase, conditioned on $(x, a)$, REVISEVAL deploys a LLM reviser $\mathcal{R}$ to revise $y$ to generate a response-adapted reference $r^\star$. Secondly, in the **evaluation** phase, taking the $(x, a, y)$ and generated $r^\star$ as input, REVISEVAL adopts LLM-as-a-Judge $\mathcal{F}_E$ to assess $y$ using $r^\star$ as the reference. Besides, we further expand REVISEVAL to support traditional reference-based metrics $\mathcal{F}_M$. The evaluation objective is to ensure that the automated evaluations align closely with human evaluations, which is introduced in Appendix G.

## 3.1 RESPONSE-ADAPTED REFERENCE GENERATION

LLMs have already demonstrated their surprising revision capabilities in various tasks, including improving specific attribution (*e.g.*, *writing style* and *grammar*) of passages (Gao et al., 2023), correcting hallucination (Akyurek et al., 2023), post-editing the generated story (Yang et al., 2022) and generating higher-quality revised responses to complement preference pairs for DPO (Guo et al., 2024; Yoon et al., 2024; Jiang et al., 2024b; Xu et al., 2024). Thus, unlike previous works treating LLMs as *discriminators* (Hu et al., 2024), we leverage the revision capabilities of LLMs to unlock the *generative* potential to offer richer and more valuable insights for evaluation. REVISEVAL deploys LLMs to revise the response $y$ from the instruction $x$ on a evaluation rubric $a$,

$$r^\star = \mathcal{R}(y|x, a), \tag{1}$$

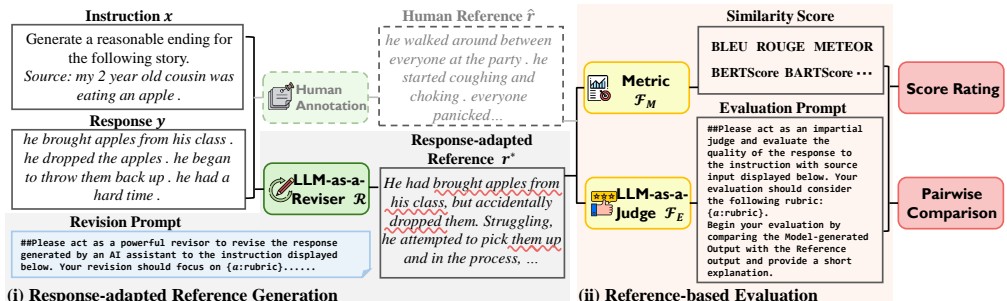

Figure 2: Illustration of our proposed **REVISEVAL**. Given an instance $(x, y, a)$, we use REVISEVAL to assess $y$ in rubric $a$. In REVISEVAL, (i) reviser generates a response-adapted reference $r^\star$ by revising the $y$ to enhance the (ii) following LLM-as-a-Judge, even classic text metrics. Here, ... represents retained segments during the generating response-adapted reference process.

where $r^\star$ is the generated response-adapted reference for the subsequent evaluation. Notably, when REVISEVAL must give the preference on two responses, $y_1$ and $y_2$, in pairwise comparison, we randomly select one for primary text to be revised while using the other as *revision guidance*. We remind reviser that this *revision guidance* may not be perfect and should be used with caution in the revision prompt,

$$r^\star = \begin{cases} \mathcal{R}(y_1|y_2, x, a) & \text{if } \mathcal{C} = 1 \\ \mathcal{R}(y_2|y_1, x, a) & \text{if } \mathcal{C} = 2 \end{cases} \tag{2}$$

where $\mathcal{C}$ is a random variable that decides which response is chosen for revision. $C$ can be either 1 or 2, with each having an equal chance of occurring. Notably, we discuss other possible revision strategies in Appendix I.2, which are less effective comparably. Introducing a qualified reference can reliably guide the evaluation process, for instance, acting as an "anchor" to reduce biases. Our strategy, which incorporates revision guidance and randomly sampling one as the primary text for revising, further reinforces fairness. This will be validated in Sec. 4.2 and 4.5.

### 3.2 REFERENCE-BASED EVALUATION

In the evaluation phase, REVISEVAL supports LLM-as-a-Judge $\mathcal{F}_E$ in a reference-based setting and remains compatible with previous metrics $\mathcal{F}_M$.

**LLM-as-a-Judge.** With powerful generalization capabilities, LLMs can serve as discriminators for evaluation, referred to as LLM-as-a-Judge. In this case, the evaluation can be operated using $\mathcal{F}_E$,

$$s = \mathcal{F}_E(y|x, a, r^\star). \tag{3}$$

This can be easily accomplished by simply using a general prompt for inference, where we clarify the instruction, response, response-adapted reference, and evaluation rubric in the prompt, as shown in Appendix A. Moreover, we implement this for open-source LLMs through finetuning evaluation data in this task format, with the detailed process provided in Appendix C.

**Metrics.** Once we get the reference $r^\star$, we can also implement classic metrics, regardless of statistical n-gram or model-based metrics. The evaluation score $s$ is:

$$s = \mathcal{F}_M(y, r^\star|[x, a]). \tag{4}$$

More specifically, when $\mathcal{F}_M$ are n-gram metrics, *e.g.*, BLEU, ROUGE, and METEOR, we can directly compute the similarity between $y$ and $r^\star$; when $\mathcal{F}_M$ are model-based metrics, *e.g.*, BERTScore and BARTScore, we can optionally input $x$ and $a$ to the metrics. The effectiveness of metrics heavily relies on the reference. When the response-adapted reference is appropriate, even a simple metric can revive its evaluation functionality in open-ended tasks. We validate this point in Sec. 4.3 and 4.4.

Table 1: Kendall ($\tau$) and Spearman ($\rho$) correlation results comparing reference-free, reference-based, and REVISEVAL methods across natural language generation tasks. This table demonstrates that, without human-annotated references, our proposed REVISEVAL substantially outperforms reference-free and reference-based methods involving both open-source and proprietary LLM-as-a-Judge.

| Methods | SUMMARIZATION $\tau$/$\rho$ | TRANSLATION $\tau$/$\rho$ | DATA2TEXT $\tau$/$\rho$ | STORY GENERATION $\tau$/$\rho$ | Avg. $\tau$/$\rho$ |
|---|---|---|---|---|---|
| *N-gram Metrics* | | | | | |
| **BLEU** | 10.66/14.42 | 14.50/19.73 | 23.13/33.29 | -1.93/-2.70 | 11.59/16.19 |
| **ROUGE** | 10.81/14.85 | 13.19/17.83 | 24.74/35.49 | -1.53/2.34 | 11.80/17.63 |
| **METEOR** | 12.37/16.72 | 16.52/18.80 | 25.58/36.27 | -1.87/-2.65 | 13.15/17.29 |
| *Model-based Metrics* | | | | | |
| **BERTScore** | 17.50/23.83 | 31.57/42.41 | 30.74/43.75 | 16.00/23.79 | 23.95/33.45 |
| **BARTScore** | 29.12/35.50 | 7.01/12.83 | 22.32/34.33 | 14.15/33.48 | 18.15/29.04 |
| **UniEval** | **35.89/47.52** | 16.08/21.90 | 28.56/38.38 | **31.22/44.46** | 27.94/38.07 |
| **GPTScore** | 28.20/37.41 | 6.50/8.90 | 19.81/28.82 | 16.36/23.91 | 17.72/24.76 |
| **InstructScore-7B** | 20.86/38.68 | 40.44/**50.43** | 30.21/38.54 | 13.50/16.13 | 26.25/35.94 |
| **TIGERScore-7B** | 28.79/35.11 | 33.65/41.50 | 32.44/42.39 | 29.72/39.26 | 31.15/39.56 |
| **Llama-3.1 8B-Inst** | 27.49/31.02 | 19.59/23.54 | 28.46/36.24 | 26.13/29.97 | 25.42/30.19 |
| *Open-Source LLM-as-a-Judge* | | | | | |
| **Ref-Free** | 27.83/31.89 | 30.84/38.66 | 38.75/49.32 | 25.74/31.72 | 30.79/37.90 |
| **Ref-Based** | 34.09/39.53 | 35.76/41.12 | **39.24**/50.87 | 8.79/10.44 | 29.47/35.49 |
| **REVISEVAL (Ours)** | 32.41/37.73 | 33.14/39.66 | 39.02/49.92 | 25.95/32.11 | **32.63/39.86** |
| *Proprietary LLM-as-a-Judge* | | | | | |
| **Ref-Free** | 31.82/38.98 | 34.62/43.38 | 37.99/49.50 | 23.81/33.29 | 32.06/41.29 |
| **Ref-Based** | 32.56/40.01 | **41.47**/45.29 | 37.35/49.02 | 17.58/24.86 | 32.24/39.80 |
| **REVISEVAL (Ours)** | 33.63/41.15 | 40.72/45.32 | 37.90/**50.93** | 25.11/35.26 | **34.34/43.17** |

# 4 EXPERIMENTS

In this section, we first present the comprehensive experimental settings in Sec. 4.1 and evaluate the REVISEVAL in LLM-as-a-Judge across various tasks in Sec. 4.2; we then verify the effectiveness of classic text evaluation metrics when using response-adapted references in Sec. 4.3; building on above findings, we compare two evaluation paradigms, combining LLM-as-a-reviser with multiple metrics and LLM-as-a-Judge, when using weak LLM in Sec. 4.4; finally, we conduct detailed comparative analysis of REVISEVAL in Sec. 4.5, such as bias reduction, inference cost and reference relevance.

## 4.1 EVALUATION SETTINGS

**Evaluation benchmarks.** We evaluate our approach on multiple classic NLG benchmarks by measuring the correlation between the evaluators/metrics and human annotations in a **scoring rating** task. We follow the experimental setting of Jiang et al. (2024a) and select four representative NLG tasks and corresponding benchmarks: **Data-to-Text** (WebNLG), **Machine Translation** (WMT-22 (zh-en)), **Text Summarization** (SummEval), and **Story Generation** (OpenMEVA), and Table 10 shows the details of these benchmarks. Additionally, we test our approach on the more challenging open-ended instruction-following benchmarks (**MT-Bench**, **Alpacafarm**, and **LLMBar**), which primarily rely on **pairwise comparison** task. Unlike the NLG benchmarks, these preference benchmarks contain more general instructions covering a broader range of tasks with more diverse responses and use accuracy to measure the evaluation performance. Remarkably, in NLG tasks, "ref-based" evaluation relies on human-annotated references as its foundation. In contrast, open-ended instruction-following tasks lack pre-existing references, so "ref-based" evaluation uses machine-generated responses as references to conduct ablation studies verifying the effectiveness of ours references.

**Base LLMs and metrics.** Our proposed REVISEVAL aims to improve evaluation performance across both LLM-as-a-Judge and classic metrics, offering enhanced results. For **proprietary LLMs**, we adopt GPT-4 as the base model and focus on implementing our paradigm during the **inference** stage. For **open-source LLMs**, we implement our method via **finetuning** the Llama 3.1-8B model. Following the Jiang et al. (2024a)'s setting on open-source models, we distill the evaluation outputs generated by GPT-4 when inputting *task instructions and corresponding evaluated responses*, and tune them in our models. Notably, our training data has no overlap with evaluation benchmarks. For

Table 2: Accuracy of LLM-as-a-Judge on instruction-following preference tasks. Our proposed REVISEVAL considerably enhances the performance of both open-source and proprietary LLM-as-a-Judge across various general evaluation tasks. Here, D.R. denotes Direct Response to instruction.

| Methods | # of Training Samples | MT-BENCH | ALPACAFARM | LLMBAR | Avg. |
|---|---|---|---|---|---|
| Open-Source LLM-as-a-Judge | | | | | |
| **JudgeLM-7B** (Zhu et al., 2023) | 100,000 | 64.1 | 53.9 | 36.3 | 51.4 |
| **PandaLM-7B** (Wang et al., 2024c) | 300,000 | 75.0 | 54.9 | 31.7 | 53.9 |
| **Auto-J-13B** (Li et al., 2024a) | 4,396 | 75.2 | 64.6 | 36.0 | 58.6 |
| **Prometheus-7B** (Kim et al., 2024a) | 100,000 | 52.8 | 33.5 | 30.1 | 38.8 |
| **Prometheus-2-7B** (Kim et al., 2024b) | 300,000 | 55.0 | 37.3 | 26.3 | 39.5 |
| **Llama 3.1-8B-Tuned** | | | | | |
| –**Ref-Free** | 9,800 | 67.4 | 61.1 | 51.1 | 59.9 |
| –**Ref-Based (Llama-D.R.)** | 9,800 | 74.9 | 61.5 | 58.9 | 65.1 |
| –**Ref-Based (GPT-4-D.R.)** | 9,800 | 78.0 | 65.5 | 63.0 | 68.8 |
| –**REVISEVAL (Llama-as-a-Reviser)** | 9,800 | 75.2 | 64.7 | 57.8 | 65.9 |
| –**REVISEVAL (GPT-4-as-a-Reviser)** | 9,800 | 79.3 | 67.1 | 64.9 | 70.4 |
| Proprietary LLM-as-a-Judge (GPT-4) | | | | | |
| **Ref-Free** | - | 81.2 | 70.9 | 72.6 | 74.9 |
| **Ref-Based (GPT-4-D.R.)** | - | 81.5 | 67.7 | **79.9** | 76.4 |
| **REVISEVAL (GPT-4-as-a-Reviser)** | - | **83.0** | **72.9** | 79.0 | **78.1** |

**classic metrics**, we cover various metrics, like *n-gram based* BLEU, ROUGE, METEOR and *model based* BERTScore, MOVERScore (Zhao et al., 2019), BARTScore, which rely heavily on references. We validate our method by assessing the utility of **reference** texts.

We ensure that our approach maintains versatility and fairness across models, with further details on prompts, inference, finetuning, and baselines in the Appendix A, B, C, D, E and F.

## 4.2 ENHANCING LLM-AS-A-JUDGE PERFORMANCE

We present the main results of our proposed REVISEVAL across NLG evaluation tasks and instruction-following preference benchmarks in Table 1 and 2. We summarize the conclusions below.

**REVISEVAL achieves stronger performance across various NLG tasks.** For the powerful proprietary LLMs, REVISEVAL outperforms reference-free and human-annotated reference-based evaluation across tasks with approximately 0.02 in Kendall correlation on average, as demonstrated in Table 1. Notably, in story generation, high-quality human references hinder evaluation for LLM-as-a-Judge, which decreases by 0.06 compared to the reference-free method in the Kendall correlation. In contrast, REVISEVAL shows that generated response-adapted reference can still greatly enhance evaluation by about 0.08 in Kendall than human reference-based evaluation. An exception is machine translation, where REVISEVAL aligns closely with reference-based methods, and we analyze this result exists a consistent rationale about reference relevance in Sec 4.5.

For the open-source LLMs, REVISEVAL not only outperforms the reference-free method but also beats the reference-based methods by over 0.03 on average in Kendall correlation. Especially in story generation, the reference-based approach is consistent with the above conclusion, trailing by approximately 0.17 in Kendall compared to our REVISEVAL. Furthermore, the LLM with specialized finetuning also performs better than the LLM with general instruction finetuning (*i.e.*, *Llama 3.1-8B Inst*) on NLG evaluation tasks, leading by about 0.02 in Kendall.

**REVISEVAL excels on open-ended instruction-following preference benchmarks.** As shown in Table 2, whether implemented on open-source or proprietary models, REVISEVAL consistently surpasses all baselines by at least 6.3% on average. The details of these baselines are listed in Appendix F, and they are tuned with tens of thousands of data for the LLM-as-a-Judge. On the same base LLM, REVISEVAL exceeds reference-free evaluation by 3%-6%. We compare REVISEVAL to reference-based evaluations followed by Zheng et al. (2023)'s setup, whose reference is the LLM's direct response to the instruction. Our approach performs better than reference-based evaluation on average when both references are generated by the same base LLMs. Furthermore, using GPT-4 as the reviser boosts Llama 3.1-8B-as-a-Judge by over 4.5% compared to Llama-as-a-Reviser, highlighting the importance of reference quality.

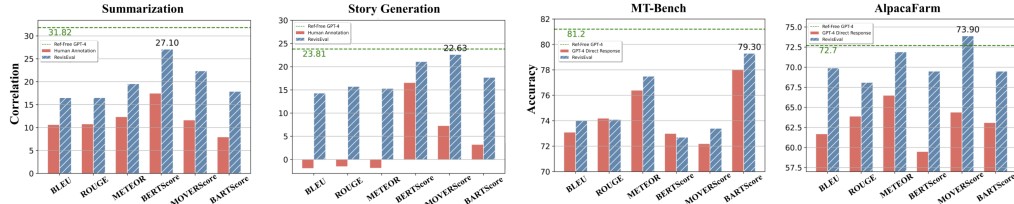

Figure 3: Comparative analysis of reference-based metrics performance using references generated by HUMAN/GPT-4 and REVISEVAL on NLG and instruction following benchmarks. REVISEVAL greatly enhance traditional ref-based metrics, even achieving them comparable to GPT-4-as-a-Judge.

LLM-as-a-Judge can be biased toward longer, verbose answers (Saito et al., 2023; Dubois et al., 2024) or answers that match a similar format (Huang et al., 2024). LLMBar (Zeng et al., 2024) is a challenging benchmark to meta-evaluate such superficial quality biases. As demonstrated in Table 2, these baselines, even after tuning $> 100K$ samples, struggle to exceed 50% accuracy, exposing bias challenge. By using response-adapted references from REVISEVAL, the weak open-source LLM-as-a-Judge's performance improves substantially by about 6%, showing **REVISEVAL can address superficial quality bias**. On proprietary LLM, REVISEVAL achieves a 3.2% improvement in accuracy compared to reference-free evaluation. Our proposed response-adapted references perform slightly worse than direct responses from the same model. This is likely because LLMBar emphasizes "instruction-following precision," where a single golden response exists for the instruction.

In summary, REVISEVAL consistently outperforms traditional ref-free and ref-based methods, and our revision provides guidance for LLM evaluation by utilizing the generative advantages of LLM.

## 4.3 INVESTIGATION ABOUT RESPONSE-ADAPTED REFERENCES

**Revision Quality.** As a post-editing mechanism to enhance text quality, we expect the revision to correct errors within the text, including subtle errors (*e.g.*, grammar), as well as higher-level issues (*e.g.*, instruction following precision and factual/logical correctness). LLMBar is an appropriate testbed for conducting detailed case studies and quality assessments. We directly use LLM to score the quality of the responses to be evaluated and the corresponding response-adapted references, focusing on correctness. Table 3 demonstrates that our response-adapted references effectively improve response accuracies. Additionally, the case studies are presented in Appendix J.1; we test REVISEVAL on JudgeBench (Tan et al., 2024) in Appendix K, focusing on factual/logical correctness and on RewardBench (Lambert et al., 2024) in Appendix N, for more challenging domains.

Table 3: Mean rating for responses and generated references in LLMBar. The detailed rating is in Table 14. Here, Responses 1 and 2 correspond to the position of the response in each pair, and the rating range is in [1, 5].

| Text | Mean Rating |
|------|-------------|
| Response 1 | 3.19 |
| Response 2 | 3.25 |
| References (Ours) | 4.75 |

**References Effectiveness.** Traditional reference-based metrics rely heavily on references, directly validating whether the references are effective. Therefore, we use response-adapted references with these metrics to observe evaluation performance across NLG and instruction-following benchmarks. We extend classic reference-based metrics to support pairwise comparisons by using an indicator function to determine preference between $y_1$ and $y_2$ based on their respective metric scores. We use human references in NLG tasks and GPT-4 direct responses in instruction-following benchmarks as the baseline references, comparing them with ours. As shown in Figure 3, REVISEVAL enables each classic metric to substantially surpass its performance based on baseline references across all tasks, particularly in more complex open-ended tasks like story generation and AlpacaFarm. Additionally, using references generated by GPT-4 as-a-Reviser combined with classic metrics can yield comparable evaluation performance to GPT-4's reference-free evaluation. Furthermore, we test response-adapted references in a multi-reference setting, with results detailed in Appendix L and M.

Table 4: Comparative analysis of weak **LLM-as-a-Judge** and weak **LLM-as-a-Reviser+classic metrics** on instruction-following tasks. Under the same finetuning training resources, a weak LLM-as-a-Reviser combined with classic metrics can produce better results.

| Metrics | MT-BENCH | ALPACAFARM | LLMBAR | Avg. |
|---|---|---|---|---|
| **LLM-as-a-Reviser** | | | | |
| **BLEU** | 64.5 | 63.9 | 51.6 | 60.0 |
| **ROUGE** | 62.0 | 63.5 | 51.6 | 59.0 |
| **METEOR** | 66.4 | 67.7 | 46.3 | 60.1 |
| **BERTScore** | 62.3 | 62.3 | **54.4** | 59.7 |
| **MOVERScore** | 61.5 | 68.3 | 51.6 | 60.5 |
| **BARTScore** | 66.9 | 61.9 | 51.3 | 60.0 |
| **MAJORITY VOTING** | 63.4 | **68.5** | 52.5 | **61.4** |
| **LLM-as-a-Judge** | **67.4** | 61.1 | 51.1 | 59.9 |

## 4.4 POTENTIAL EVALUATION PARADIGM FOR WEAK LARGE LANGUAGE MODELS

We observe that 1) in the Sec. 4.2, all weak LLMs still exhibit a notable gap compared to GPT-4-as-a-Judge, even after extensive training with high-quality, evaluation-specific data, and 2) in the Sec. 4.3, classic metrics combined with response-adapted references generated by GPT-4 can achieve performance close to the reference-free GPT-4-as-a-Judge. Thus, *should we consider a potential evaluation paradigm of "weak LLM-as-a-Reviser + metrics" instead of "weak LLMs-as-a-Judge"?*

To explore this, we compare "Llama-as-a-Judge" with "Llama-as-a-Reviser + metrics," as shown in Table 4. When using references generated by "Llama-as-a-Reviser", we find that BLEU, METEOR, MOVERScore, and BARTScore can surpass "Llama-as-a-Judge" on average across 3 tasks. Furthermore, we apply a *majority voting* across multiple metrics, outperforming "Llama-as-a-Judge" over 1.5% on average. This suggests that instead of continuously training weak LLMs to improve their evaluative **discrimination** capabilities, leveraging their **generation** abilities for revision may be more effective. Without extra inference costs, this approach can lead to better evaluation outcomes. We offer a general guideline: majority voting is recommended as it provides a stable superior evaluation approach, recognizing that no single metric outperforms others across all benchmarks.

Table 5: Positional bias analysis in pair comparison evaluations when applying different evaluation paradigms. This table presents the ratio of changed evaluation results after swapping the response position. A lower proportion indicates less positional bias. REVISEVAL stands out as the best, exhibiting the lowest bias among all paradigms.

| Paradigms | MT-BENCH | | ALPACAFARM | | LLMBAR | |
|---|---|---|---|---|---|---|
| | LLAMA 3.1-8B | GPT-4 | LLAMA 3.1-8B | GPT-4 | LLAMA 3.1-8B | GPT-4 |
| **Ref-Free** | 49.1 | 10.3 | 61.1 | 20.0 | 44.6 | 17.9 |
| **Ref-Based** | 22.8 | 6.5 | 34.1 | 22.2 | 32.5 | 11.2 |
| **REVISEVAL** | **20.5** | **5.9** | **30.1** | 19.9 | 30.3 | **7.9** |

## 4.5 COMPARATIVE ANALYSIS TO OTHER EVALUATION PARADIGMS

**Positional bias analysis.** Positional bias (Wang et al., 2024a; Zheng et al., 2023; Wu & Aji, 2023; Chen et al., 2024) occurs when human or LLM evaluators tend to favor one side in a pairwise comparison, regardless of answer quality. We investigate this bias by swapping answer positions and taking the LLM to re-evaluate, as shown in Table 5. The results indicate that reference-based evaluation decisions have less variation than reference-free ones. REVISEVAL is generally 2%-4% lower on the reference-based evaluation, showing better consistency. This result is probably because REVISEVAL provides references more closely aligned with the answers, further minimizing bias.

**The impact analysis of inference cost.** Compared to the reference-free approach, REVISEVAL requires two cycles of inference (*i.e.*, revision and evaluation). To show the impact of extra inference cost, we further conduct three cycles of reference-free evaluations using extra different temperatures (0.3 and 0.7), followed by majority voting (see Table 6). For GPT-4, additional evaluation cycles

Table 6: Ablation study on the impact of inference cost. Increasing evaluation cycles to match or exceed REVISEVAL's inference cost in reference-free did not improve accuracy. This shows that REVISEVAL's superior performance is not from twice inference.

| Inference Cost | MT-BENCH | | ALPACAFARM | | LLMBAR | |
|---|---|---|---|---|---|---|
| | LLAMA 3.1-8B | GPT-4 | LLAMA 3.1-8B | GPT-4 | LLAMA 3.1-8B | GPT-4 |
| **1-Cycle Ref-Free** | 67.4 | 81.2 | 61.1 | 70.9 | 51.1 | 72.6 |
| **3-Cycle Ref-Free** | 64.1 | 81.2 | 54.9 | 71.9 | 53.2 | 74.9 |
| **REVISEVAL (2-Cycle)** | **71.3** | **83.0** | **64.7** | **72.9** | **54.9** | **79.0** |

Table 7: Comparative analysis of how reference-based evaluation effectiveness varies with changes in the similarity between the response and reference texts across different constructing reference strategies. Here, Effectiveness, $\mathcal{P}_{\text{ref}}/\mathcal{P}_{\text{free}}$, refers to the performance ratio between reference-based and reference-free evaluation, where $\mathcal{P}$ denotes the evaluation performance, *e.g.*, Acc and Corr; similarity is still measured by BERTScore.

| | Reference Source | WMT-22(EN-ZH) | WEBNLG | MT-BENCH | SUMMEVAL | ALPACAFARM | ROC |
|---|---|---|---|---|---|---|---|
| **Similarity** | Human/GPT-4 | **65.12** | 59.09 | 25.00 | 23.51 | 13.04 | 12.86 |
| | RevisEval | 63.03 (-3.3%) | **76.41** (+29.3%) | **30.29** (+21.2%) | **35.63** (+51.6%) | **35.72** (+173.9%) | **27.57** (+114.4%) |
| **Effectiveness** | Human/GPT-4 | **1.20** | 0.98 | 1.00 | 1.02 | 0.95 | 0.74 |
| | RevisEval | 1.18 (-0.02) | **1.00** (+0.02) | **1.02** (+0.02) | **1.06** (+0.04) | **1.03** (+0.08) | **1.05** (+0.31) |

slightly improve accuracy but still lag behind ours by 1%-4%. For weak LLMs, more cycles led to worse performance. The above results indicate that our approach provides more valuable guidance.

**Evaluation performance improves with increased relevance between reference and responses.** It's been observed that the less relevant a reference is to the response, the less effective it is for evaluation in previous work and Figure 1. We further verify whether this trend holds true with our method. We define effectiveness to describe whether reference-based evaluation is more effective than reference-free evaluation. As shown in Table 7, the increasing similarity between reference and evaluated responses generally leads to better evaluation effectiveness. This explains why our method doesn't perform as well in translation, where human references are already highly similar to the response. For other tasks, human or GPT-4 direct-response references have lower similarity than references of REVISEVAL, leading to a lower effectiveness. Additionally, for different tasks, the similarity between the human/GPT-4 reference and the evaluated response varies, reflecting the open-ended generative degree of this task. A lower similarity indicates a greater diversity of potential valid responses. In this context, as the task becomes more open-ended, the effectiveness of REVISEVAL shows a greater improvement than the human/GPT-4 reference.

# 5   CASE STUDY: HOW DOES IT WORK?

We show two representative examples in Table 8, one from the instruction-following task (Alpaca-Farm) and the other from the NLG evaluation task (story generation). The reference-free evaluation tends to be influenced by verbose text, often leading to incorrect judgments, *e.g.*, "provides a clearer answer...". While the LLM's direct response or human reference might not be erroneous and can fulfill the instructions well, for instance, "*Happy and Joyful belong to the word group of synonyms, ...,*" in the first case, it has no relevance to responses to be evaluated and resulting "neither assistant directly addressed the classification," judgment. In contrast, the reference generated by REVISEVAL is a more adaptive guidance, "*Emotions*", accurately helping the LLM to align the crucial elements, "the clarity and completeness". This is also evident in the second case, where differences between the response and the revised text, such as "had brought," "accidentally dropped," and others, directly highlight *fluency* issues with the response. In other words, this demonstrates a transparent potential. Furthermore, we provide evaluation discrepancy statistics between REVISEVAL and other evaluations to observe how different evaluation methods, as demonstrated in Appendix I.3.

Table 8: An instruction-following open-ended case of pairwise comparison and a story-generation case of score rating evaluated by different paradigms. Score/Preference refers to accurate evaluation result and Score/Preference refers to wrong result. We emphasize the noisy evaluation elements in red, while highlighting the useful elements of the evaluation in green. Here, the base LLM is GPT-4.

| |
|---|
| **Instruction**: Classify the following to the correct word group. \n\n Happy, Joyful |
| **Response 1**             **Response 2**
Happy -> Emotions            Emotions |
| **Reference-free Evaluation**:
   Both Assistant A and Assistant B . . . . . . Both responses are correct, but one might argue that including the words with the categorization, as Assistant A does, provides a clearer answer and specifically indicates that both words have been classified. . . . . .
   - Final Verdict: [[A]] |
| **Referfence-based Evaluation**:
   *Direct Response-as-Reference*: Happy and Joyful belong to the word group of synonyms, as they both express similar feelings of pleasure and contentment.

   . . . . . . Assistant A, while correct in stating that "Happy" belongs to "Emotions," did not mention "Joyful" or classify them together as synonyms. Rather, it only assigned a category to one word.
   Assistant B . . . . . but fails to classify both words as instructed, missing the relationship between "Happy" and "Joyful."
   Neither assistant directly addressed the classification as "synonyms," which was central to the user's instruction.
   - Final Verdict: [[A]] |
| **REVISEVAL Evaluation**:
   *Response-adapted Reference*: Emotions

   . . . . . . Assistant A . . . . . . However, it is clear from the classification provided that Assistant A understands the context and correctly categorizes "Happy" but overlooks "Joyful".
   Assistant B simply responds with "Emotions", effectively covering both "Happy" and "Joyful". This response uses a more concise and generalized approach, which aligns perfectly with the reference answer.
   Based on the reference answer and the clarity and completeness in addressing both terms, Assistant B's response is superior. . . . . . .
   - Final Verdict: [[B]] |
| **Instruction**: Generate a reasonable ending for the following story. \n Source Input: my 2 year old cousin was eating an apple . |
| **Response**: he brought apples from his class. he dropped the apples. he began to throw them back up. he had a hard time.
**Gold Score**: 3.8 |
| **Reference-free Evaluation**:
   . . . . . . but it lacks coherence and logical flow. The sentences seem disconnected, and the events described do not build upon each other in a meaningful way. The output does not demonstrate a clear understanding of the story's context or the character's actions.
   - Rating: [[1]] |
| **Referfence-based Evaluation**:
   *Human Reference*: he walked around between everyone at the party . he started coughing and choking . everyone panicked and started hitting him on the back . he eventually spit out the apple piece .

   The model-generated output is not fluent and lacks coherence in comparison to the reference output.
   . . . . . . . . . . .
   Overall, the model-generated output fails to meet the fluency criteria due to its lack of coherence and grammatical errors.
   - Rating: [[1]] |
| **REVISEVAL Evaluation**:
   *Response-adapted Reference*: He had brought apples from his class, but accidentally dropped them. Struggling, he attempted to pick them up and in the process, began tossing them around. He was having a hard time managing the apples.

   The model-generated output is somewhat fluent but lacks coherence and completeness in comparison to the reference output. . . . . . The reference output provides a more detailed and logical sequence of events, enhancing the narrative's coherence and clarity.
   . . . . . .
   Overall, the model-generated output is fluent but could benefit from improvements in coherence and detail to match the quality of the reference output.
   - Rating: [[3]] |

# 6 CONCLUSION

We introduce a novel evaluation paradigm that utilizes LLMs' revision capabilities to generate response-adapted references, enhancing LLM-as-a-Judge's reliability. The references produced by REVISEVAL significantly improve even simple n-gram metrics, achieving performance on par with LLM-as-a-Judge. This is particularly beneficial for weaker LLMs, which often struggle to improve despite extensive training, offering an efficient way to enhance evaluation capabilities despite limited training. Our findings emphasize (1) the underestimated importance of references and (2) the potential of LLMs' generative strengths to improve evaluation by increasing reference relevance. Looking ahead, REVISEVAL enables promising extensions: (i) integrating a Reviser with classic metrics to better support smaller LLMs, (ii) applying revision mechanisms to multi-modal tasks like MLLM, and (iii) incorporating REVISEVAL into multi-agent pipelines for robust evaluation frameworks.

ACKNOWLEDGEMENTS

This work was supported by the Early Career Scheme (No. CityU 21219323) and the General Research Fund (No. CityU 11220324) of the University Grants Committee (UGC), and the NSFC Young Scientists Fund (No. 9240127).

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

```
Please act as an impartial judge and evaluate the quality of the responses
provided by two AI assistants to the user question displayed below. You should
choose the assistant that follows the user's instructions and answers the user's
question better. Your evaluation should consider factors such as {rubric: the
helpfulness, relevance, accuracy, depth, creativity, and level of detail of their
responses}. Begin your evaluation by comparing the two responses and provide a
short explanation. Avoid any position biases and ensure that the order in which
the responses were presented does not influence your decision. Do not allow the
length of the responses to influence your evaluation. Do not favor certain names
of the assistants. Be as objective as possible. After providing your explanation,
output your final verdict by strictly following this format: "[[A]]" if assistant
A is better, "[[B]]" if assistant B is better.

[Instruction]
{instruction}

[The Start of Assistant A's Answer]
{response_output_1}
[The End of Assistant A's Answer]

[The Start of Assistant B's Answer]
{response_output_2}
[The End of Assistant B's Answer]
```

Figure 4: The prompt of reference-free pairwise comparison evaluation.

```
Please act as an impartial judge and evaluate the quality of the responses
provided by two AI assistants to the user question displayed below. You should
choose the assistant that follows the user's instructions and answers the user's
question better. Your evaluation should consider factors such as the {rubric:
helpfulness, relevance, accuracy, depth, creativity, and level of detail of their
responses}. And I also give a reliable reference answer, and begin your
evaluation by comparing the two responses with the reference answer and provide a
short explanation. Avoid any position biases and ensure that the order in which
the responses were presented does not influence your decision. Do not allow the
length of the responses to influence your evaluation. Do not favor certain names
of the assistants. Be as objective as possible. After providing your explanation,
output your final verdict by strictly following this format: "[[A]]" if assistant
A is better, "[[B]]" if assistant B is better.

[Instruction]
{instruction}

[Reference Answer]
{ref_answer}

[The Start of Assistant A's Answer]
{response_1}
[The End of Assistant A's Answer]

[The Start of Assistant B's Answer]
{response_2}
[The End of Assistant B's Answer]
```

Figure 5: The prompt of reference-based pairwise comparison evaluation.

## A  PROMPT TEMPLATE

We provide the prompt templates used for evaluation and revision. These prompts are either taken directly from MT-Bench or adapted from it, ensuring the universality of our proposed paradigm.

```
Please act as an impartial judge and evaluate the quality of the model-generated
output provided by an AI assistant to the instruction with source input displayed
below. Your evaluation should consider the following aspect: {rubric}.
Begin your evaluation by providing a short explanation. Be as objective as
possible. After providing your explanation, please rate the response on a scale
of 1 to 5 by strictly following this format: "[[rating]]", for example: "Rating:
[[3]]".

Instruction: {instruction}
{input_context}

Model-generated Output: {response_output}
```

Figure 6: The prompt of reference-free score rating evaluation.

```
Please act as an impartial judge and evaluate the quality of the response
generated by an AI assistant to the instruction with source input displayed below.
Your evaluation should consider the following aspect: {rubric}.
Begin your evaluation by comparing the Model-generated Output with the Reference
output and provide a short explanation. Be as objective as possible. After
providing your explanation, please rate the response on a scale of 1 to 5 by
strictly following this format: "[[rating]]", for example: "Rating: [[3]]".

Instruction: {instruction}
{input_context}

Model-generated Output: {response_output}

Reference output: {ref_output}
```

Figure 7: The prompt of reference-based score rating evaluation.

```
Please act as a powerful reviser to revise the response generated by an AI
assistant to the instruction displayed below. You should revise the response to
follow the user's instructions and answer the user's instruction better. Your
revision should consider factors such as the {rubric}. If the original response
is good enough, simply output the original answer.

**Instruction:**{instruction}

**Model-Generated Response:**{response_output_1}

I also give you another model-generated answer, which is not necessarily of
better quality, as a reference for your revision, and you can draw on its
strengths and avoid its weaknesses.
**Another Answer:**{response_output_2}

Do NOT provide any explanation for your response.
ONLY output the complete revised answer without saying anything else.
```

Figure 8: The prompt of LLM-as-a-reviser for pairwise comparison.

```
Please act as a powerful reviser to revise the response generated by an AI
assistant to the instruction and source input displayed below. Please revise this
output to be more {rubric}. If model-generated output is already good enough,
simply output that original output.
**Instruction:**{instruction}
{input_context}

**Model-generated Output:**{response_output}

Do NOT provide any explanation for your response.
ONLY output the complete revised answer without saying anything else.
```

Figure 9: The prompt of LLM-as-a-reviser for score rating.

```
{instruction}
```

Figure 10: The prompt of LLM direct response to instruction.

Table 9: The Statistics of NLG Evaluation Training Data.

| Task | Aspects | Samples Items | Evaluation Items |
|---|---|---|---|
| Summarization | fluency,consistency,coherence,relevance | 2886 | 11544 |
| Translation | accuracy | 6000 | 6000 |
| Data2Text | accuracy,fluency | 3098 | 6196 |
| Story Generation | fluency,consistency,style matching | 1052 | 3156 |

## B    INFERENCE SETTING FOR PROPRIETARY MODEL

**Base Model.**  We choose GPT-4 as the base model for our evaluation and revision. For reproducibility, we used the GPT-4 version GPT-4-TURBO-2024-04-09, with a temperature setting of 0.0.

## C    FINETUNING SETTING FOR OPEN-SOURCED MODEL

**Base Model.**  We choose LLAMA 3.1-8B-INST [1] as the base model for our evaluation and revision. Here, we want to clarify that we choose the INSTRUCT model as the base model because finetuning on this model yields better evaluation and revision results than the PRETRAINED model.

**Training Setting.**  We followed the common setup for supervised instruction finetuning, with a *context length* $= 2048$, *epochs* $= 3$, *batch size* $= 128$, and *learning rate* $= 2e - 5$.

**Distilling Setting.**  Whether finetuning open-source models for evaluation or revision capabilities, the training data comes from the generation of a powerful model prompted by the same data source. The model we choose to distill is still GPT-4, with the same version and inference settings as mentioned above.

**Distilling Data Source.**  We depict the distilling data flow in Figure 11. For NLG evaluation, ideally, we would have a variety of erroneous samples along with human evaluation scores for them. However, such data typically exists only in test sets, making it unavailable for training and often limited in quantity. Therefore, we choose MetricInstruct [2], proposed by Jiang et al. (2024a), as our

---

[1] https://huggingface.co/meta-llama/Meta-Llama-3.1-8B-Instruct
[2] https://huggingface.co/datasets/TIGER-Lab/MetricInstruct

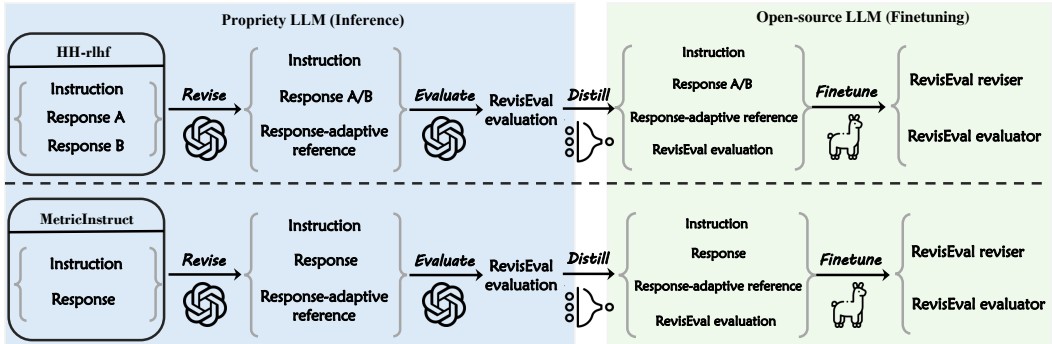

Figure 11: In our data distillation process for open-source LLMs, we utilize HH-rlhf and MetricIn-struct as the primary data sources. We then employ a proprietary LLM to perform RevisEval, generating both the revisions and corresponding evaluation outputs. Finally, we fine-tune the open-source LLM using this enriched dataset.

training data source. This dataset provides a large volume of diverse erroneous texts, which serve as the basis for evaluation. From the $40K+$ data points, we filter out other NLG tasks and apply our previously mentioned prompts with corresponding aspects, maintaining the same inference settings to generate evaluation scores and reasoning for these error samples. Detailed statistics are presented in the Table 9. Although the overall dataset size is relatively small compared to other works specifically designed to train evaluators, the NLG evaluation data we assess remains held-out from these training samples.

Unlike NLG evaluation data that lacks human-labeled evaluation, preference data typically contains substantial human-labeled preference annotations without evaluation. We choose the most commonly used hh-rlhf [3] (Bai et al., 2022) dataset, applying the aforementioned prompts and inference settings to conduct evaluations on this data to get the evaluation. We select the preference correctness intersection of reference-free evaluation, reference-based evaluation, and gold preference annotations, ensuring both accuracy and fairness when comparing the performance of LLMs under different evaluation methods post-training. In the end, we selected $10,000$ samples, each containing corresponding revisions, reference-free evaluation, and reference-based evaluation. These three sets were then used to train the same model separately, ensuring that no new information is introduced to alter the distribution.

**Finetuning LLM-as-a-Judge adaptive to NLG.** Each instance in filtered MetricInstruct is a tuple (response, input context, task instruction, aspect), then use GPT-4 to get the corresponding (reference-free evaluation, revision text, REVEVO evaluation) to each instance. Subsequently, we separately use these data to finetune LLM to get the reference-free evaluator, LLM-as-a-reviser, and reference-based evaluator.

Each instance in filtered hh-rlhf is a tuple (instruction, response a, response b), then use GPT-4 to get the corresponding (reference-free evaluation, revision text, REVEVO evaluation) to each instance. Subsequently, we separately use these data to finetune LLM to get the reference-free evaluator, LLM-as-a-reviser, and reference-based evaluator.

**Decoding Setting.** When the finetuned model executes the evaluation and revision tasks, the de-coding setting uses a greedy decoding strategy with a *max output length* $= 1024$ and *temperature* $= 0.01$.

Table 10: The Statistics of NLG Evaluation Benchmarks.

| Task | Benchmark | Response Source | Inputs Items | Samples Items |
|------|-----------|-----------------|--------------|---------------|
| Summarization | SummEval (Fabbri et al., 2021) | 16 Models | 100 | 1600 |
| Translation | WMT-22 (zh-en) (Kocmi et al., 2022) | 18 Models | 1875 | 33750 |
| Data2Text | WebNLG-2020 (Zhou & Lampouras, 2020) | 18 Models | 179 | 2848 |
| Story Generation | OpenMEVA (ROC) (Guan et al., 2021) | 5 Models | 200 | 1000 |

## D    BENCHMARKS

### D.1    NLG EVALUATION BENCHMARKS

Traditional text generation tasks and their corresponding evaluation benchmarks are highly diverse. Based on the varying degrees of freeform in text generation tasks, we select four representative **Machine Translation**, **Data-to-Text**, **Summarization**, and **Story Generation**. We follow the experiment setting of Tigerscore and choose specific benchmarks for each task, and their statistics are shown in the Table 10.

### D.2    INSTRUCTION FOLLOWING PREFERENCE BENCHMARKS

With their powerful generalization capabilities, LLMs have become the focus of research on NLP generation abilities. Evaluating LLMs requires more challenging tasks that can assess their generalization capabilities. Here, we selected three representative benchmarks: MT-BENCH (*abbr.*, MT-BENCH_HUMAN-JUDGEMENT), ALPACAFARM, and LLMBAR.

**MT-Bench.** This dataset comprises $3.3K$ expert-level pairwise human evaluations of model responses, generated by six models across 80 MT-Bench questions. The six models include GPT-4, GPT-3.5, CLAUDE-V1, VICUNA-13B, ALPACA-13B, and LLAMA-13B, offering a diverse representation of powerful language models. The topic of subtasks is consisted of *Writing*, *Roleplay*, *Reasoning Math*, *Coding*, *Extraction*, *STEM* and *Humanities*. MT-Bench is the most common benchmark for evaluating LLM-as-a-Judge, and our results validate the feasibility of our experiments. We select the first round of dialogues from this dataset as the evaluation data, containing 1284 cases.

**AlpacaFarm.** We utilize HUMAN-CROSSANNOTATION [4] set specifically designed for evaluating the reliability of evaluators, following the ALPACAFARM process. Each instance in this dataset contains cross-annotations from 4 human experts. Additionally, the tasks in this dataset are more diverse, open-ended, and challenging, making the preference annotations more reliable. Notably, since four experts conduct the preference annotations, some instances resulted in ties, where two annotators favored the first option and the other two favored the second. We filtered out these tied cases, leaving a final evaluation dataset of 501 instances.

**LLMBar.** LLMBar is a meta-evaluation benchmark designed to test how well LLM evaluators can identify instruction-following outputs. It consists of two parts: (1) The Natural set, which gathers instances from existing human-preference datasets, filtered and adjusted to ensure a clear preference for each instance. (2) The Adversarial set, where the authors intentionally create misleading outputs that superficially seem good but deviate from the instructions, to challenge the evaluators. The Natural set measures performance in real-world conditions, while the Adversarial set tests evaluators' ability to detect true instruction-following. The overall size is 419.

## E    BASELINES IN NLG EVALUATION TASKS

**N-gram Metrics.** N-gram text generation metrics are commonly used to evaluate the text quality generated by models, especially in tasks like machine translation and summarization. While these metrics are simple and efficient, they come with notable limitations: they are highly sensitive to

---

[3] https://huggingface.co/datasets/Anthropic/hh-rlhf
[4] https://huggingface.co/datasets/tatsu-lab/alpaca_eval/blob/main/alpaca_farm_human_crossannotations.json

Table 11: The Lists of weak LLM-as-a-Judge.

| Model | Base Model | Instruction | Response Annotation | Evaluation Scheme | Training Samples |
|---|---|---|---|---|---|
| JudgeLM | Vicuna-7B | Alpaca-GPT4, Dolly-15K... | 11 models (Alpaca, Vicuna...) | GPT-4 Pairwise Grading | $100K$ |
| PandaLM | LLaMA-7B | Alpaca 52K | 5 models (LLaMA, Bloom...) | GPT3.5 Pairwise Selection | $300K$ |
| Auto-J | LLaMA2-13B-chat | Chatbot Arena, OpenAI WebGPT... | Preference Datasets | Human Pairwise Selection, Pointwise Grading | 4396 |
| Prometheus | LLaMA2-7B-chat | GPT-4 Generated | GPT-4 Generated | GPT-4 Pointwise Grading | $100K$ |
| Prometheus-2 | Mistral-7B-v2.0 | GPT-4 Generated | GPT-4 Generated | GPT-4 Pointwise Grading Preference | $300K$ |

surface-level differences, such as word order or vocabulary choice, which may fail to capture the true meaning or fluency of the generated text. In our evaluation, we use the most widely adopted metrics: BLEU, ROUGE-L, and METEOR.

**Model-based Metrics.** To capture the semantic-level meaning of generated text, researchers have started using models like BERT and BART as the foundation for text evaluation. Typical examples include BERTScore, BARTScore, and Moverscore. BERTScore computes the similarity between two text sequences based on the contextual embeddings from BERT, while Moverscore enhances this by adding many-to-one alignment. BARTScore, on the other hand, uses BART to calculate the probability of converting the response to the reference text as a score. All of these are reference-based model metrics.

Additionally, there are reference-free model-based metrics. These metrics are trained on specific task datasets, allowing the model to internalize relevant information. As a result, the model can generate evaluations without needing reference texts and transform them into scores. For instance, UNIEVAL uses data augmentation to expand a task-specific dataset to 30K examples and fine-tunes on T5, which is why it performs exceptionally well in summarization tasks.

**LLM-as-a-Judge.** Following the exciting advancements in large language models (LLMs), the most straightforward approach has been to replace the models in previous model-based metrics, such as BART, with larger models. GPTScore follows this concept, and despite its simplicity, it delivers notable results. Moreover, leveraging the vast internal knowledge of open-source LLMs, more powerful and interpretable evaluators can be developed, such as INSTRUCTScore and TIGERScore.

Xu et al. (2023) argue that performing error analysis on given reference texts enhances evaluation explainability and reliability. They used NLG evaluation data and employed GPT-4 to perform error-based assessments. These outputs were then paired with the response to train an Llama model, resulting in a training dataset of 10K examples. TIGERScore goes a step further by proposing a reference-free approach. Using a similar strategy, they collected 40K data points for training, masking the reference text during the process.

## F BASELINES IN INSTRUCTION FOLLOWING PREFERENCE BENCHMARKS

**LLM-as-a-Judges.** The baseline models are listed in the Table 11. Considering scalability and cost, researchers have long sought to achieve evaluation performance on weaker LLMs that is comparable to that of stronger LLMs. The most straightforward approach to this challenge has been to automatically generate more preference-related data, and many of these efforts have followed this strategy.

**N-gram Metrics & Model-based Metrics.** Unlike the NLG-Evaluation benchmark, the LLM-as-Judge benchmark has largely moved away from using n-gram metrics and model-based metrics. This shift is due to several characteristics of the test samples in such benchmarks: (1) The response space is extremely large and unconstrained, making reference annotations both unhelpful and prohibitively expensive; (2) These metrics do not perform well for tasks such as coding or math, where even models like BERT struggle to capture semantic-level meaning. In this study, we apply these metrics

Table 12: Pearson correlation coefficients comparing non-reference, static-reference, and dynamic-reference methods across various text generation tasks and Instruction-Following Benchmarks. This table summarizes the performance of these methods in generating summarization, translation, data-to-text, and story-generation tasks.

| Methods | SUMMARIZATION | TRANSLATION | DATA2TEXT | STORY GENERATION | Avg. |
|---|---|---|---|---|---|
| n-gram Metrics | | | | | |
| **BLEU** | 14.13 | 17.47 | 34.29 | -3.89 | 15.50 |
| **ROUGE** | 15.36 | 16.26 | 35.85 | -0.22 | 16.81 |
| **METEOR** | 18.69 | 18.80 | 36.30 | -1.02 | 18.19 |
| Reference-free Metrics | | | | | |
| **BERTScore** | 26.26 | 37.65 | 48.22 | 26.58 | 34.68 |
| **BARTScore** | 19.73 | 29.04 | 47.89 | 17.76 | 28.61 |
| **UniEval** | **53.22** | 23.11 | 51.14 | 44.88 | 43.09 |
| **GPTScore** | 13.47 | 21.05 | 48.70 | 18.94 | 25.54 |
| **InstructScore-7B** | 27.40 | 51.55 | 47.28 | 12.81 | 34.76 |
| **TIGERScore-7B** | 43.95 | 37.70 | 49.13 | 39.90 | 42.67 |
| **Llama-3.1 8B-Instruct** | 25.89 | 27.84 | 31.15 | 31.04 | 28.98 |
| Open-Sourced LLM-as-a-Judge | | | | | |
| **Ref-Free** | 33.61 | 25.14 | 53.36 | 35.02 | 36.78 |
| **Ref-Based** | 42.32 | 29.99 | 48.52 | 11.92 | 33.19 |
| **REVEVO(Ours)** | 39.69 | 29.58 | 53.87 | 29.04 | 38.05 |
| Proprietary LLM-as-a-Judge | | | | | |
| **Ref-Free** | 42.12 | 41.35 | **54.26** | 33.50 | 43.56 |
| **Ref-Based** | 43.31 | **44.11** | 53.98 | 24.63 | 41.51 |
| **REVEVO(Ours)** | 43.81 | 43.92 | 54.25 | **35.07** | **44.26** |

to demonstrate that, when reference texts are highly relevant, these otherwise "inapplicable" metrics can be reactivated and produce meaningful results.

## G  META EVALUATION

Meta-evaluation aims to assess the performance of automated metrics by measuring how well the automated evaluations $y_{\text{auto}}$ align with human evaluations $y_{\text{human}}$. For score ratings, we calculate the correlation values across all $N$ samples, represented as:

$$Corr = g\left([y_{\text{auto}}^1, \ldots, y_{\text{auto}}^n], [y_{\text{human}}^1, \ldots, y_{\text{human}}^n]\right),$$

where $g$ can adopt various correlation functions, *e.g.*, Spearman. For pair-wise comparison evaluations, accuracy is typically used as the evaluation metric,

$$Acc = \frac{1}{|N|} \sum_{(i, o_1^i, o_2^i) \in N} \mathbb{I}[y_{\text{auto}}^i = p^i]$$

**NLG Tasks.** In the NLG evaluation task, it's crucial to assess various rubrics of the text during the evaluation process. For the four selected tasks, we've outlined the specific rubrics to be evaluated in the accompanying Table 9. In both the SummEval and Story-Generation tasks, we evaluate multiple rubrics independently, calculating the correlation coefficient for each one. Subsequently, we compute the average correlation coefficient across all rubrics to obtain an overall assessment for each task. This comprehensive approach ensures a more nuanced and accurate evaluation of the model's performance across different dimensions.

## H  PEARSON CORRELATION IN NLG EVALUATION TASKS

We also supplement the Pearson Correlation results in the NLG Evaluation Tasks.

Table 13: Statistics on the differential proportion between REVISEVAL and each of the other two evaluation methods. A higher ratio indicates a greater evaluation difference between the two mechanisms.

| Comparative Evaluation | MT-BENCH | ALPACAFARM | LLMBAR |
|---|---|---|---|
| **Ref-Free** | 8.1 | 22.8 | 16.9 |
| **Ref-Based (GPT-4 Direct Response)** | 9.7 | 21.6 | 16.2 |

# I  OTHER ANALYSIS

## I.1  THE BENEFITS OF TRAINING RESOURCE SCALE FOR LLM-AS-A-JUDGE ARE QUESTIONABLE.

As shown in Table 2 and 11, Despite being trained with extensive evaluation-specific resources, these LLM-as-a-Judge baselines fail to achieve evaluation performance comparable to GPT-4, particularly on the adversarially designed LLMBar, where they perform worse than random selection. While substantial effort is put into designing and generating a large amount of training data for these LLMs, the results are even less effective than our evaluator trained on just 10,000 samples from the hh-rlhf dataset. The possible reasons for this could be: 1. The inherent capabilities of the base model play a more crucial role; 2. Simply increasing the volume of training data does not yield significant benefits; 3. Efforts should be focused on other potentials to enhance the evaluator, such as the method we propose.

## I.2  OTHER REVISION STRATEGIES FOR PAIRWISE COMPARISON

When revising two given responses to generate a reference, we experiment with two different revision strategies. In our work, we adopt a strategy where one text is randomly selected as the primary text to be revised, while the other serves as the revision guidance. In addition to this, we try the following two prompt strategies: a) Revising a single text based on both responses. b) Revising each response separately. However, the outcomes of these two strategies are unsatisfactory. Strategy (a) exhibit a tendency to forcibly merge the two responses during the revision process, resulting in a generated text that lacked logical consistency. Strategy (b), on the other hand, lead to a revised text with very low similarity to the other response, inevitably favoring the one chosen for revision. Neither method is capable of producing a satisfactory reference. We anticipate that future research might provide more revision strategies or approaches to more effectively combine multiple responses and generate a high-quality reference. We look forward to seeing further developments in this area.

## I.3  EVALUATION DIFFERENTIAL OF DIFFERENT METHODS

We analyse the differential between REVISEVAL and two other evaluation methods, showcasing the proportion of samples where the evaluation decisions differed. A higher proportion indicates a greater difference in the evaluation mechanisms of the two methods. The Table 13 present that REVISEVAL and the other two evaluation methods make different decisions on 22% of the samples in AlpacaFarm and 16% of the samples in LLMbar. This indicates that our evaluation method has a significant difference in mechanism-level, compared to the other two methods. Furthermore, it suggests aggregating these differing decisions could potentially lead to a more reliable final evaluation.

# J  THE CORRECTNESS STUDY OF RESPONSE-ADAPTED RESPONSE

Revision is a post-edit modification method aimed at improving the quality of the response. The ability to correct subtle errors in the text is a key factor we observe to determine the effectiveness of the revision. Experiments on LLMBar and BLEU/ROUGE provide relevant evidence to support this to some degrees: i) LLMBar Benchmark: the performance of the judge on LLMBar relies on its ability to detect subtle errors in instruction following. As shown in Table 2, REVISEVAL enhances the LLM-as-a-Judge's performance on LLMBar, demonstrating that our references effectively validate these errors. ii) BLEU/ROUGE: as N-gram metrics, completely depend on references to identify subtle errors. In Sec. 4.3 and 4.4, we input the references generated by REVISEVAL into BLEU/ROUGE to

evaluate their effectiveness. By computing accuracy and the correlation to human scores, we show that our references successfully capture and validate these errors. Although our existing method has demonstrated advantages in both LLMBar and classic n-gram metrics, we will provide a more direct study and analysis here.

## J.1 CASE STUDY

On LLMBar, we present response samples modified by REVISEVAL.

> **Question**: Convert from celsius to fahrenheit. Temperature in Celsius: 15
>
> **Response**: 77˘b0F
>
> **Response-Adapted Reference**: 59˘b0F

> **Question**: Sort them in lexicographical order: nnapples, grapes, bananas, oranges, kiwi, pears, plums, oranges, cherries, apples, tangerines, peaches
>
> **Response**: apples, bananas, cherries, grapes, kiwi, oranges, peaches, pears, plums, tangerines
>
> **Response-Adapted Reference**: apples, apples, bananas, cherries, grapes, kiwi, oranges, oranges, peaches, pears, plums, tangerines

We can observe that compared with the response, the adapted reference contains fewer subtle errors. This provides the guidance to make llm-as-a-judge easier to evaluate, as containing errors directly affects the quality of the response, leading to a higher alignment with human evaluators than reference-free evaluators.

## J.2 DIRECT QUALITY EVALUATION ON RESPONSE-ADAPTED REFERENCES

We directly score the quality (1-5) of the adapted references on the correctness aspect compared to the original responses using LLM-as-a-Judge.

Table 14: Comparasive Correctness Rating to the original responses and response-adapted references on LLMBar. Here, we directly use LLM-as-a-Judge (GPT-4o) to rate the responses and references on correctness.

| | ADVERSARIAL NEIGHBOR | ADVERSARIAL GPTINST | ADVERSARIAL GPTOUT | ADVERSARIAL MANUAL | NATURAL | OVERALL |
|---|---|---|---|---|---|---|
| **Response 1** | 3.27 | 3.03 | 3.47 | 3.39 | 2.99 | 3.19 |
| **Response 2** | 3.16 | 3.01 | 3.32 | 3.47 | 3.44 | 3.25 |
| **Response-adapted Reference** | 4.72 | 4.73 | 4.53 | 4.91 | 4.82 | 4.75 |

In this Tab. 14, Response 1 and Response 2 refer to the position of this response in the pairs. On 5 subsets of LLMBar, the references (revised by REVISEVAL-gpt-4-turbo) have consistently better correctness than the responses, which means REVISEVAL detects and corrects the subtle errors.

## K EVALUATION FOCUSING ON FACTUAL ACCURACY

While historical work and our experiments have validated that revision is an effective approach for improving text generation, in this section, we further examine whether revision remains effective in modifying text errors in factual correctness.

First, previous studies (Pan et al., 2024; Gao et al., 2023) have demonstrated that revision can be applied to correct factual errors, such as hallucination. Here, we provide a more direct experiment: using JudgeBench (Tan et al., 2024), a benchmark for LLM-as-a-Judge to directly evaluate factual errors in model-generated text, we test whether REVISEVAL can improve the effectiveness of LLM-as-a-Judge in this context.

Table 15: Comparative Performance on JudgeBench. Here, the metric is Accuracy.

| METHOD | KNOLWEDGE | MATH | REASONING | CODING | OVERALL |
|---|---|---|---|---|---|
| **LLM-as-a-Judge (gpt-4-turbo)** | 48.05 | 69.64 | 56.12 | 38.09 | 52.57 |
| **RevisEval (gpt-4-turbo)** | 64.29 | 64.29 | 70.41 | 45.24 | 63.71 |
| **LLM-as-a-Judge (gpt-4o)** | 53.2 | 55.4 | 49.0 | 35.7 | 50.3 |
| **RevisEval (gpt-4o)** | 72.7 | 58.9 | 66.3 | 33.3 | 64.0 |
| **LLM-as-a-Judge (gpt-4o-mini)** | 64.3 | 62.5 | 65.3 | 42.9 | 61.7 |
| **RevisEval (gpt-4o)** | 70.8 | 64.3 | 63.3 | 54.8 | 65.7 |

Here, JudgeBench includes multiple samples about factual correctness across various domains, and REVISEVAL demonstrates strong performance in evaluating factual correctness on different domains in Tab. 15.

## L    MULTIPLE REFINED-GRAINED REFERENCES

In the traditional NLG-Evaluation Tasks, *e.g.*, Machine Translation (Freitag et al., 2020), multiple human-annotated references have been proven to be a simple yet effective method to improve the reliability of evaluators/metrics. And, Tang et al. (2024) further uses LLMs to paraphrase the pre-existing human-annotated references to generate multiple references for the following evaluation. Based on this, we hope to verify whether REVISEVAL works for multiple references and more refined-grained references, and further extend the applicability of REVISEVAL.

In the original REVISEVAL setting, we revise the response once to generate one reference, wherein the reviser prompt, *"Your revision should consider factors such as the helpfulness, relevance, accuracy, depth, creativity, and level of detail of their responses"*, we include all aspects in one prompt. Now, we conduct separate fine-grained revisions to generate multiple references, where each reviser prompt only includes one aspect (helpful, accuracy, relevance, depth, creativity), then use them in the following evaluation separately. For each case, we will evaluate it based on each reference, and we will get multiple predicted preferences or scores. For the preference evaluation task, we will run a majority-voting to get a final preference; for the score-rating evaluation task, we will obtain a mean score. We name this new evaluation pipeline as FINE-GRAINED REVISEVAL, and test its accuracy in the MT-Bench as follows,

Table 16: Performance comparison of evaluation methods on MT-Bench. The results show that FINE-GRAINED REVISEVAL, which incorporates multiple refined-grained references, further improves evaluation accuracy over REVISEVAL and LLM-as-a-Judge, demonstrating the effectiveness of this extended strategy.

| METHOD | GPT-4-TURBO | GPT-4O-MINI |
|---|---|---|
| **LLM-as-a-Judge** | 81.18 | 80.29 |
| **RevisEval** | 83.01 | 81.38 |
| **Finegrained-RevisEval** | 84.13 | 81.99 |

As presented in this Tab. 16, REVISEVAL has been improved in the multiple refine-grained references setting, indicating it works for this classic strategy.

## M    COMPARISON WITH DIV-REF

Both DIV-REF (Tang et al., 2024) and ours leverage references to enhance evaluation performance. However, there are some differences between the two methods:

i) Methodology: DIV-REF diversifies pre-labeling references (through paraphrasing), so it cannot support reference-free benchmarks, *e.g.*, MT-Bench, AlpacaFarm. and this work also only tests in NLG-tasks. In contrast, we find that the reason for the ineffectiveness of pre-existing references is

lacking relevance to the response, as shown in Fig 1. Therefore, REVISEVAL proposes to revise the response and create a post-generated reference with higher relevance towards itself than pre-existing ones. Additionally, REVISEVAL supports reference-free benchmarks, while DIV-REF can not.

ii) Performance: To comprehensively validate the effectiveness of the references generated by both mechanisms, we employ classic N-gram metrics (ROUGE), model-based metrics (BERTScore and MoverScore), and an LLM evaluator (GPT-4-Turbo, aligned with our version) for testing.

For Tang et al. (2024)'s method, we diversify a pre-existing human-labeled reference ten times to produce ten references, followed by running the reference-based metric separately to get 10 scores and calculating the mean score. For REVISEVAL (ours), we do not rely on human references; instead, we revise the response to generate one response-adapted reference. We then conduct the reference-based metric to obtain a score. For each specific aspect (*e.g.*, fluency), we compute the correlation between the predicted score and human evaluation scores, then average the correlation values across these aspects to derive the final performance score for this benchmark. We compare two methods in SummEval and Story Generation, and the correlation we choose Spearman, the results are as below:

Table 17: Comparison of DIV-REF and REVISEVAL on SummEval and Story Generation tasks. REVISEVAL achieves superior results in most cases, particularly in story generation, highlighting its advantage in addressing the limitations of pre-existing references.

| Methods | SummEval | | | | Story Generation | | | |
|---|---|---|---|---|---|---|---|---|
| | ROUGE | BERTScore | MOVERScore | GPT-4-Turbo | ROUGE | BERTScore | MOVERScore | GPT-4-Turbo |
| Human-Reference | 14.85 | 23.83 | 19.73 | 40.01 | 2.34 | 23.79 | 16.47 | 24.86 |
| DIV-REF | 18.25 | 28.13 | 23.47 | 43.82 | 1.53 | 25.79 | 15.48 | 27.38 |
| REVISEVAL | 19.65 | 29.47 | 25.85 | 41.15 | 17.24 | 25.84 | 26.89 | 35.26 |

As we can observe in the Tab. 17, both DIV-REF and our REVISEVAL outperform human references, indicating the effectiveness of both methods. Comparably, our method is the best-performed one in most cases, indicating its effectiveness, especially in story generation. In summary, DIV-REF is a piece of solid evidence of the ineffectiveness of pre-existing references. However, REVISEVAL differs from DIV-REF in how we address such ineffectiveness and whether supporting a reference-free benchmark. We will incorporate this comparative experiment in the next version.

# N  PERFORMANCE ON REWARDBENCH

While we have tested on LLMBar, a challenging enough benchmark, to verify this issue, Reward-Bench (Lambert et al., 2024) is the latest popular challenging benchmark covering more challenging domains, such as CHAT-HARD, REASONING, and SAFETY subsets. So, we decide to choose the RewardBench as an extensive experiment, the result is below,

Table 18: Comparative Performance on RewardBench.

| METHOD | CHAT | CHAT-HARD | SAFETY | REASONING | OVERALL |
|---|---|---|---|---|---|
| LLM-as-a-Judge (gpt-4-turbo) | 97.76 | 80.04 | 88.51 | 90.01 | 89.04 |
| RevisEval (gpt-4-turbo) | 97.76 | 80.04 | 88.51 | 90.01 | 89.04 |
| LLM-as-a-Judge (gpt-4o) | 98.60 | 79.17 | 92.03 | 94.47 | 91.96 |
| RevisEval (gpt-4o) | 97.76 | 83.55 | 93.51 | 95.53 | 93.47 |
| LLM-as-a-Judge (gpt-4o-mini) | 96.37 | 60.09 | 91.89 | 81.21 | 82.45 |
| RevisEval (gpt-4o) | 93.30 | 65.13 | 93.08 | 86.35 | 85.63 |

REVISEVAL can improve LLM-as-a-Judge consistently on RewardBench. Especially in challenging subsets, REVISEVAL surpasses LLM-as-a-Judge stably.

## O   PERFORMANCE ON CHALLENGING REASONING BENCHMARKS

To further explore the potential boundaries of our method, extremely challenging reasoning-based generation benchmarks serve as an excellent testbed. We introduce two new benchmarks, GPQA (Rein et al., 2024) and Omni (Gao et al., 2024a), both of which are state-of-the-art and highly challenging in the field of reasoning.

However, the initial setups of these benchmarks were not designed for evaluating LLM-as-a-Judge. Specifically, they lack positive-negative response pairs for the judge to discriminate between. To address this, we modify the benchmark setups by using LLMs to generate positive-negative response pairs for each question.

- **Negative responses** are relatively easy to generate. We used GPT-4o to repeatedly sample and produce incorrect responses.
- **Positive responses**, being harder to create due to the complexity of the questions, were generated through correcting the negative responses with the oracle solution-as-references by GPT-4o.

As a result, we obtain two responses with similar styles but differing levels of accuracy.

In this experiment, we adapt GPQA and Omni to test the reliability of LLM-as-a-Judge. The evaluation task assesses whether LLM-as-a-Judge can more accurately select the positive response without access to the oracle answer. We introduce two baselines for comparison, 1) **Vanilla**: LLM-as-a-Judge directly selects the better response without any reference; 2) **Ref-based**: LLM answers the question first, and its response is used as the reference.

Table 19: Comparison of LLM-as-a-Judge Baselines and REVISEVAL on GPQA and Omni benchmarks.

| Methods | GPQA | | | | Omni |
|---|---|---|---|---|---|
| | GPQA_EXTENDED | GPQA_MAIN | GPQA_DIAMOND | OVERALL | |
| **Vanilla(gpt-4o-mini)** | 53.11 | 54.69 | 58.59 | 54.61 | 51.99 |
| **Ref-based(gpt-4o-mini)** | 52.01 | 50.89 | 54.05 | 51.93 | 47.38 |
| **RevisEval(gpt-4o-mini)** | 54.21 | 53.79 | 58.59 | 54.78 | 53.00 |
| **Vanilla(gpt-4o)** | 67.39 | 71.21 | 70.71 | 69.39 | 60.86 |
| **Ref-based(gpt-4o)** | 59.89 | 60.26 | 61.62 | 60.32 | 57.99 |
| **RevisEval(gpt-4o)** | 69.23 | 70.98 | 75.76 | 70.91 | 61.43 |
| **Vanilla(gpt-4-turbo** | 70.33 | 66.52 | 68.68 | 68.62 | 61.11 |
| **Ref-based(gpt-4-turbo)** | 54.58 | 57.14 | 56.57 | 55.79 | 59.01 |
| **RevisEval(gpt-4-turbo)** | 71.06 | 67.19 | 68.69 | 69.21 | 62.37 |

We can draw the following conclusions from the Table 19: i) In the absence of an Oracle solution, our method remains effective even on extremely challenging benchmarks. ii) Compared to having the LLM directly answer the question, REVISEVAL provides a more effective solution for generating references with LLMs.

## P   JUSTIFICATION OF USING REVISION AS A RELIABLE METHOD TO ENHANCE TEXT QUALITY

The revision process in LLMs is not a simple copy operation. We emphasize that our revision mechanism incorporates information from two responses rather than solely one response being revised (shown in Sec. 3). Due to the training objective of LLMs—*LLMs are predominantly trained to generate high-quality, human-like outputs, with limited exposure to the distribution of low-quality or flawed data*—the probability of retaining correct segments is significantly higher than that of retaining erroneous segments during the revision process. This observation is also utilized by numerous studies (Yang et al., 2022; Akyurek et al., 2023; Guo et al., 2024; Gao et al., 2023) that post-revision by LLMs is a reliable mechanism to improve response quality.

For example, let $E$ represent erroneous segments in the response, and $H$ represent high-quality segments. Consider two responses: $R_1$: $E_1, H_2, H_3$; $R_2$: $H_2, H_3, H_4$ and $R_2$ has a higher quality. After revision, the reference $R^\star$ is likely to retain $H_2, H_3, H_4$ while eliminating $E_1$. Then LLM-as-a-Judge will pick the $R_2$, as $R^\star$ would align more with high-quality segments overall. If the LLM fails to remove and retains it, the resulting $R^\star = E_1, H_2, H_3, H_4$ would be equally closer to $R_1$ and $R_2$. Then, inputting the reference adapted from $R_1$ into LLM-as-a-Judge would make no difference from LLM-as-a-Judge without reference.

