# OpenReview forum: "RevisEval: Improving LLM-as-a-Judge via Response-Adapted References"
_ICLR.cc/2025/Conference — ICLR 2025 Poster_

### Official Review · Reviewer_zE3r · 2024-11-02

**Soundness:** 2
**Presentation:** 2
**Contribution:** 2
**Rating:** 5
**Confidence:** 4

**Summary:**

The paper proposes a simple solution to enhance reference-based evaluation of LLM-as-a-Judges. Instead of using pre-made references, the work introduces a novel evaluation paradigm, "Revise-and-Evaluation," where an LLM revises the provided input to generate a reference answer. The authors note that this method is effective in creating a reference similar to the response in terms of style and different artifacts, effectively accounting for the quality of the answer only. The methodology can be expanded to classical evaluation methodologies like BLEU, ROUGE, and more. The methodology is tested on diverse benchmarks.

**Strengths:**

1. The paper proposes a simple and straightforward solution to improve reference-based evaluation. The methodology is easy to implement and shows promising improvement in quality on different benchmarks.

2. The methodology shows strong robustness, naturally controlling for style, and shows nice performance on adversarial benchmarks like LLM Bar, despite relatively small training.

**Weaknesses:**

Please see the questions section.

**Questions:**

1. While the improvements look promising, there are some questions about the effectiveness of the proposed solution. Recent meta-evaluation works like Reward Bench [1] show that reward models are much more powerful than LLM-as-Judges in proxying human responses. Does the proposed methodology have a benefit against RMs?

2. Automated evaluators are also widely used as a proxy for human preference in RLHF. An additional step to generate revisions makes the whole process slower and expensive. Hence, while the performance may be promising, it seems like it limits the usage of automated evaluators. Where do you expect this methodology to be used?

3. Mandating a revision step before evaluation assumes that the revising model can refine the answer better. What if the question-response pair is too difficult for the model to revise? Will the methodology still be effective?


[1] https://arxiv.org/abs/2403.13787

---

> ### Author Response · Authors · 2024-11-17
> **The Response to Reviewer zE3r (1/2)**
>
> Hi Reviewer zE3r, we appreciate your review. We hope we can resolve your confusion about our work.
>
> ---
>
> > ##### **Q1: While the improvements look promising, there are some questions about the effectiveness of the proposed solution. Recent meta-evaluation works like Reward Bench [1] show that reward models are much more powerful than LLM-as-Judges in proxying human responses. Does the proposed methodology have a benefit against RMs?**
>
> Thanks for your proposed question. We hope to clear up some confusion:
>
> i). **Our proposed method and its motivation/objective are not targeted to reward models (RMs)**.
>
> a. *Applicable Domains*: RMs provide quality signal to further optimize LLMs in the **post-training** stage [1], but LLM-as-a-Judge replaces humans in the **evaluation** stage and more domains[2],
>
> b. *Quality Expressions*: RMs provide preference signals (0/1) only[3], but LLM-as-a-Judge offers diverse forms of quality (e.g., preference and score ratings) and detailed **judgment/evaluation analysis paragraphs**;
>
> c. *Modelling*: RMs uses **Bradley-Terry modeling**, LLM-as-a-Judge uses **supervised fine-tuning and prompt engineering**;
>
> d. *Seperate Inference*: RMs only need **input response-pairs**, LLM-as-a-Judge need input **response-pairs and prefix-prompt**, as evidenced in RewardBench's GitHub generative.py and rewardbench.py.
>
> Our work target is to improve LLM-as-a-Judge rather than RMs.
>
> ii). **The reward model and LLM-as-a-Judge have comparble evaluation effectiveness in proxying human.**
>
> In Table 9 of RewardBench, there are 3 LLM-as-a-Judges and 2 RMs in the top-5 of the leaderboard. Hence, we think it is hard to directly conclude that reward models have stronger preference prediction capabilities than LLM-as-a-Judge. We think both domains are worthy of investigation.
>
> iii). **Our approach may also benefit RMs**.
>
> On the one hand, our method achieves strong preference prediction results; conversely, it demonstrates that a smaller language model combined with traditional metrics can deliver a cost-effective yet accurate evaluation. This might even suggest a new direction for RMs.
>
> We hope our response provides you with greater confidence in the effectiveness of our method.
>
> > ##### **Q2: Automated evaluators are also widely used as a proxy for human preference in RLHF. An additional step to generate revisions makes the whole process slower and expensive. Hence, while the performance may be promising, it seems like it limits the usage of automated evaluators. Where do you expect this methodology to be used?**
>
> Thank you for your valuable question regarding the cost. Firstly, We continue to emphasize that the goal of our approach is to be applied during the *evaluation* phase, rather than RLHF in the *post-training* phase. In fact, we have advantages in terms of cost and speed:
>
> i). Compared to previous works [4,5], which incur costs from multiple calls (e.g., generating 10 references), our method only requires 1 reference, resulting in lower costs.
>
> ii) We also offer a lower-cost paradigm in the paper, namely llm-as-a-reviser + classic metric. The revision process involves fewer tokens than generating a judgment, and the classic metric also incurs no cost. The combination of the two can lead to more efficient evaluation results. The inference Costs are presented in tokens per case as below:
>
> Paradigm|RewardBench|MTBench|LLMBar|
> ---|---|---|---|
> LLM-as-a-Judge|228 tokens|232 tokens|236 tokens|
> LLM-as-a-Reviser + Metric|171 tokens|209 tokens|152 tokens|
>
> As shown in this table, our work has a lower and faster inference cost. Furthermore, we have showed LLM-as-a-Reviser+Metric has a better performance than LLM-as-a-Judge in Table.3 in our paper. Therefore, RevisEval is a relatively efficient new evaluation paradigm.
>
> iii). We acknowledge that RLHF may be more concerned with costs, and our method can still provide potential benefits for the cost. For example, recent work [6,7] suggests that the reward model should also generate intermediate CoT evaluation to improve evaluation accuracy (which also validates the effectiveness of LLM-as-a-Judge), and our method can reduce this cost.

---

> ### Author Response · Authors · 2024-11-17
> **The Response to Reviewer zE3r (2/2)**
>
> > ##### **Q3: Mandating a revision step before evaluation assumes that the revising model can refine the answer better. What if the question-response pair is too difficult for the model to revise? Will the methodology still be effective?**
>
> Thank you for your question about handling difficulty. Our approach has advantages in this issue.
>
> i) LLMBar, an adversarially designed benchmark that is quite challenging in terms of instruction-following:
>
> In our manual script, our method performs better than baselines.
>
> Additionally, we further tested on other sets that are more difficult in terms of question-response difficulty.
>
> ii) Chat-Hard, Reasoning and Safety Subsets in RewardBench:
>
> We report this experiment results in the general response part. As demonstrated in #GR2, RevisEval consistently surpasses the LLM-as-a-Judge across multiple difficult samples.
>
> We will update these experiments in our next version.
>
>
> > ##### **Final Claim**
>
> LLM-as-a-Judge represents a distinct and impactful area of research with broad applications across various domains, especially as a substitute for human evaluation in assessing model generation capabilities. Our proposed RevisEval is designed to enhance the effectiveness of LLM-as-a-Judge. Exploring its potential to integrate with and improve RMs is an exciting direction for our future work.
>
> ##### **References**
> [1] Training language models to follow instructions with human feedback, In Neurips' 22
>
> [2] Judging LLM-as-a-Judge with MT-Bench and Chatbot Arena, In Neurips'23
>
> [3] A Survey of Reinforcement Learning from Human Feedback, Arxiv. 2312.14925, Citation~80
>
> [4] Not All Metrics Are Guilty: Improving NLG benchmarks by Diversifying References, In NAACL' 24
>
> [5] Reference-Guided Verdict: LLMs-as-Judges in Automatic Evaluation of Free-Form Text, Arxiv.2408.09235
>
> [6] Generative Reward Models, UnderReview in ICLR' 25
>
> [7] Generative Verifiers: Reward Modeling as Next-Token Prediction, In NeurIPS' 24 MathAI workshop
>
> ---
>
> We hope our response helps the reviewer better understand how we think about this work. Any future discussion and comments are more than welcome.

---

> > ### Comment · Reviewer_zE3r · 2024-11-19
> >
> > Thanks for the comparison between LLM-as-a-Judge and RMs. As recent RMs also leverage CoT for better performance, my concerns on the issues may be resolved. Accordingly, I have revised my scores.
> >
> > However, my concerns on Q3 persists. What if I'm trying to use the LLM-as-a-Judge to evaluate a benchmark like GPQA [1], or FrontierMath [2], where the model is likely to fail? Would the benefits still persist?
> >
> > [1] https://arxiv.org/abs/2311.12022
> > [2] https://arxiv.org/abs/2411.04872v1

---

> ### Author Response · Authors · 2024-11-20
> **The Response to Reviewer zE3r about Q3**
>
> We sincerely appreciate your acceptance of our responses in Q1 and Q2.
>
> Regarding Q3, our previous general response #GR2 has provided results for RevisEval's performance on the Chat-hard, Reasoning, and Safety benchmarks in RewardBench. Here, we extract this part to present it specifically,
>
> ||CHAT-Hard|Safety|Reasoning|
> |-|-|-|-|
> LLM-as-a-Judge(gpt-4-turbo)|80.04|88.51|90.01|
> Reviseval(gpt-4-turbo)|81.14|90.01|91.89|
> |-|-|-|-|
> LLM-as-a-Judge(gpt-4o-mini)|60.09| 91.89|81.21|
> RevisEval(gpt-4o-mini)|65.13|93.08|86.35|
> |-|-|-|-|
> LLM-as-a-Judge(gpt-4o)|79.17| 92.03|94.47|
> RevisEval(gpt-4o)|83.55|93.51|95.53|
>
> Notably, there are code generation (e.g., C, Python) and mathematical reasoning tasks in the Reasoning subset.
>
> While we achieved improved outcomes, we did not anticipate the level of difficulty you expected in your review of "too difficult."
>
> Your suggestion is both intriguing and valuable, offering a great opportunity to further examine the potential of our method. As such, we decide to test RevisEval on the two benchmarks you proposed: **GPQA** and **FrontierMath**.
>
> Unfortunately, FrontierMath is not a publicly available dataset, as indicated in its Section 2.4:
>
> -      "To minimize the risk of problems and solutions being disseminated online, we encouraged all submissions to be conducted through secure, encrypted channels." -- FrontierMath
>
> As a result, we were unable to access its data for operating. Therefore, we choose **OMNI-MATH** [1], a similarly challenging and recently released **olympiad-level mathematical** benchmark with goals aligned to FrontierMath, to conduct our tests.
>
> >#### **GPQA: graduate-level questions in subdomains of physics, chemistry, and biology**
>
> GPQA provides the question with 4 choices, 1 oracle solution, and the label (choice). For verifying the LLM-as-a-Judge (pick a better response from a pair), we use gpt-4o to output one negative solution for each question as the negative response, then we use (oracle solution, negative solution) as the pair to be judged. The goal is to evaluate if the judge can pick the oracle solution. We also use the MTBench evaluation prompt, where we modify the multiple focusing aspects to the "reasoning accuracy" aspect.
>
> ||gpqa_extended|gpqa_main|gpqa_diamond|overall|
> |-|-|-|-|-|
> |LLM-as-a-Judge(gpt-4o-mini)|10.07|12.50|13.13|11.49|
> |RevisEval(gpt-4o-mini)|19.07|20.53|18.69|19.55|
> |-|-|-|-|-|
> |LLM-as-a-Judge(gpt-4-turbo)|34.25|38.84|36.36|36.32|
> |RevisEval(gpt-4-turbo)|37.00|41.96|39.90|39.35|
> |-|-|-|-|-|
> |LLM-as-a-Judge(gpt-4o)|37.00|37.05|32.32|36.24|
> |RevisEval(gpt-4o)|39.37|39.51|34.85|38.67|
>
> Here, the metric is accuracy.
>
> >#### **Omni: A universal of olympiad level mathematic benchmarks for large language models**
>
> Omni also only provides the question with 1 oracle solution, and contains 4428 samples.  For verifying the LLM-as-a-Judge (pick a better response from a pair), we use gpt-4o to output one negative solution for each question as the negative response,  then we use (oracle solution, negative solution) as the pair to be judged. We also use the MTBench evaluation prompt, where we modify the multiple focusing aspects to the "reasoning accuracy" aspect.
>
> |Method|Acc. of Evaluation|
> |-|-|
> |LLM-as-a-Judge(gpt-4o-mini)|19.67|
> |RevisEval(gpt-4o-mini)|20.21|
> |-|-|
> |LLM-as-a-Judge(gpt-4-turbo)|37.90|
> |RevisEval(gpt-4-turbo)|38.41|
> |-|-|
> |LLM-as-a-Judge(gpt-4o)|37.15|
> |RevisEval(gpt-4o)|41.82|
>
>  In summary, these benchmarks are really challenging for both the text generation ability and evaluation ability of LLM, as demonstrated in these tables. Our method has a better evaluation performance than LLM-as-a-Judge on different subsets.
>
> We hope this verification enables us to have confidence in RevisEval.
>
> ### **References**
>
> [1] Omni-MATH: A Universal Olympiad Level Mathematic Benchmark For Large Language Models. Arxiv.2410.07985

---

> ### Author Response · Authors · 2024-11-23
> **Appreciate any further feedback**
>
> Dear Reviewer zE3r,
> As the discussion phase is coming to a close soon, we look forward to hearing from you and would greatly appreciate any further feedback you can provide. Your insights would be invaluable in helping us improve the quality of our paper. Thanks!

---

> > ### Author Response · Authors · 2024-11-28
> > **New Refined Experiment setup!**
> >
> > These days, we have redesigned a new experimental setup to further refine the experiments related to GPQA and Omni.
> > The previous setup directly uses an oracle reasoning solution as a positive response, and GPT-4o generates an incorrect response as a negative response. We observe the baseline accuracy is relatively low, the reason is that the oracle solution differs notably from natural language-style responses. In this new setting, we both use the GPT-4o's response as positive-negative pairs to ensure the base style is natural language. To generate the correct response with a similar language style, we first generate a response using GPT-4o, and then use Oracle Solution as a reference to correct the response (by GPT-4o). In this process, it will remain the original natural language style but correct the logical errors.
> >
> > In this new setup, we adapt GPQA and Omni into experiments aimed at improving the reliability of LLM-as-a-Judge. The evaluation task tests whether LLM-as-a-Judge can more accurately select the positive response without access to the Oracle answer.
> >
> > We introduce two baselines for comparison:
> >
> > Vanilla: LLM-as-a-Judge directly selects the better response without any reference.
> >
> > Ref-based: LLM first answers the question, and its response is used as the reference.
> >
> > 1. GPQA
> >
> > |Method|gpqa_extended|gpqa_main|gpqa_diamond|overall|
> > | - | - | - | - | - |
> > |Vanilla(gpt-4o-mini)|53.11|54.69|58.59|54.61|
> > |Ref-based(gpt-4o-mini)|52.01|50.89|54.05|51.93|
> > | RevisEval(gpt-4o-mini) | 54.21 | 53.79 | 58.59 | 54.78 |
> > | - | - | - | - | - |
> > | Vanilla(gpt-4o)        | 67.39 | 71.21 | 70.71 | 69.39 |
> > | Ref-based(gpt-4o)      | 59.89 | 60.26 | 61.62 | 60.32 |
> > | RevisEval(gpt-4o)      | 69.23 | 70.98 | 75.76 | 70.91 |
> > | - | - | - | - | - |
> > | Vanilla(gpt-4turbo     | 70.33 | 66.52 | 68.68 | 68.62 |
> > | Ref-based(gpt-4turbo)  | 54.58 | 57.14 | 56.57 | 55.79 |
> > | RevisEval(gpt-4-turbo) | 71.06 | 67.19 | 68.69 | 69.21 |
> >
> > 2. Omni
> >
> > |-|Accuracy|
> > |-|-|
> > |Vanilla(gpt-4o-mini)|51.99|
> > |Ref-based(gpt-4o-mini)|47.38|
> > |RevisEval(gpt-4o-mini)|53.00|
> > |-|-|-|-|-|
> > |Vanilla(gpt-4o)|60.86|
> > |Ref-based(gpt-4o)|57.99|
> > |RevisEval(gpt-4o)|61.43|
> > |-|-|-|-|-|
> > |Vanilla(gpt-4turbo|61.11|
> > |Ref-based(gpt-4turbo)|59.01|
> > |RevisEval(gpt-4-turbo)|62.37|
> >
> > We can draw the following conclusions:
> >
> > 1. In the absence of an Oracle solution, our method remains effective even on extremely challenging benchmarks.
> > 2. Compared to having the LLM directly answer the question, RevisEval provides a more effective solution for generating references with LLMs.
> >
> > We will include both experiments in the revised manuscript.

---

> > > ### Author Response · Authors · 2024-12-04
> > >
> > > We sincerely appreciate your previous feedback and look forward to your new insights, which have greatly enhanced the quality and clarity of our work. As we finalize our revisions, we remain eager and hopeful for your recognition of our rearticulated contributions. Our work represents a fresh rethinking of the effectiveness of references in evaluation, and we hope it inspires further discussion and exploration in this area, guided by your valuable suggestions.

---

### Official Review · Reviewer_6W2K · 2024-11-02

**Soundness:** 4
**Presentation:** 3
**Contribution:** 2
**Rating:** 6
**Confidence:** 4

**Summary:**

The work proposes a simple and straightforward evaluation method that involves modifying and enhancing the output text to be evaluated, using it as the reference for further evaluation, motivated by the potentially unsatisfactory quality of traditional references. They experiment with various setups, including using strong and weak LLMs as revisors and employing both traditional evaluation metrics and LLM-based evaluators.

**Strengths:**

The proposed method is intuitive and reasonable, with a straightforward implementation that advances previous work using LLMs to generate references for evaluation. They also consider a comprehensive range of experimental setups, baseline methods, and evaluation benchmarks to verify the effectiveness of their method, resulting in solid experimental analyses.

**Weaknesses:**

Given that previous studies have already utilized LLMs to generate higher-quality references as replacements for traditional references (Tang et al., 2024), the innovation and contribution of this method are somewhat diminished. I believe they could further enhance the analysis by more comprehensively comparing these two approaches for generating references (generation as reference vs. revision as reference). Additionally, I suggest exploring the use of more refined response-adapted references, such as having the revisor focus on specific dimensions during evaluation, to allow for a richer and more diverse discussion.

The experiments in this work are thorough, but they may be somewhat distracting. First, the main experimental results presented in Tables 1 and 2 involve some inconsistent demonstrations; for example, both tables include the "Open-Source LLM-as-a-Judge" part, but the types of methods involved seem different. In Table 1, it’s unclear whether "Ref-Based" refers to references generated by the corresponding LLMs or the original references, which is important. And Sections 4.3 and 4.4 may not be as critical and could be moved to the appendix, given the availability of stronger evaluation methods; this would allow space for more in-depth experiments and analysis.

**Reference**

Not All Metrics Are Guilty: Improving NLG Evaluation by Diversifying References (Tang et al., NAACL 2024)

**Questions:**

Please refer to Weaknesses.

---

> ### Author Response · Authors · 2024-11-17
> **The Response to Reviewer 6W2K (1/2)**
>
> Hi Reviewer #6W2K, we appreciate your detailed review. We hope we can resolve your confusion about our work and make our paper better.
>
> ---
>
> > ##### **W1: “Given that previous studies have already utilized LLMs to generate higher-quality references as replacements for traditional references (Tang et al., 2024), the innovation and contribution of this method are somewhat diminished. I believe they could further enhance the analysis by more comprehensively comparing these two approaches for generating references (generation as reference vs. revision as reference).”**
>
> Thanks for your valuable suggestion. First, Thanks a lot for bringing up this interesting work. We will discuss the paper accordingly in future versions. Our discussion on Tang's method will include the following aspects:
>
> i). **Methodology**: Tang' s approach diversifies **pre-labeling references** (through paraphrasing), so it cannot support reference-free benchmarks, e.g., MT-Bench, AlpacaFarm. and this work also only tests in NLG-tasks. In contrast, we find that the reason for the ineffectiveness of pre-existing references is lacking relevance to the response, as shown in Fig 1. Therefore, RevisEval proposes to revise the response and create a **post-generated reference** with higher relevance towards itself than pre-existing ones. Additionally, RevisEval supports reference-free benchmarks,  while Tang' s approach can not.
>
> ii). **Evaluating Performance (Spearman) on NLG-Benchmarks**:
>
> Both Tang's work and ours leverage references to enhance evaluation performance. To comprehensively validate the effectiveness of the references generated by both mechanisms, we employ classic N-gram metrics (ROUGE), model-based metrics (BERTScore and MoverScore), and an LLM evaluator (GPT-4-Turbo, aligned with our version) for testing.
>
> For Tang's method, we diversify a pre-existing human-labeled reference ten times to produce ten references, followed by running the reference-based metric separately to get 10 scores and calculating the mean score. For RevisEval (ours), we do not rely on human references; instead, we revise the response to generate one response-adapted reference. We then conduct the reference-based metric to obtain a score. For each specific aspect (e.g., fluency), we compute the correlation between the predicted score and human evaluation scores, then average the correlation values across these aspects to derive the final performance score for this benchmark. We compare two methods in SummEval and Story Generation, and the correlation we choose Spearman, the results are as below:
>
> In SummEval,
> |Methods|ROUGE|BERTScore|MOVERScore|gpt-4-turbo|
> |---|---|---|---|---|
> |Human-Reference|14.85|23.83|19.73|40.01|
> |Tang' s Div-Ref|18.25|28.13|23.47|43.82|
> |RevisEval|19.65|29.47|25.85|41.15|
>
> In Story Generation,
>
> |Methods|ROUGE|BERTScore|MOVERScore|gpt-4-turbo|
> |---|---|---|---|---|
> |Human-Reference|2.34|23.79|16.47|24.86|
> |Tang' s Div-Ref|1.53|25.79|15.48|27.38|
> |RevisEval|17.24|25.84|26.89|35.26|
>
> As we can observe, both Tang' s method and our RevisEval outperform human references, indicating the effectiveness of both methods. Comparably, our method is the best-performed one in most of cases, indicating its effectiveness, especially in **story generation**.
>
> In summary, Tang' s method is a piece of solid evidence of the ineffectiveness of pre-existing references. However, RevisEval differs from Tang' s method in how we address such ineffectiveness and whether supporting a reference-free benchmark. We will incorporate this comparative experiment in the next version.
>
>
> > #####  **W2: “Additionally, I suggest exploring the use of more refined response-adapted references, such as having the revisor focus on specific dimensions during evaluation, to allow for a richer and more diverse discussion.”**
>
>
> We sincerely appreciate the reviewer's valuable and innovative suggestion, which provides a strong complement to further expand our approach. We give the experiment to verify it in general response. As we reported in GR1 of general responses, multiple finegrained references can help our method further improve its effectiveness. As you suggested, the Finegrained-RevisEval has a better performance, which shows a more extensive application of RevisEval.

---

> ### Author Response · Authors · 2024-11-17
> **The Response to Reviewer 6W2K (2/2)**
>
> > #####  **W3: “The experiments in this work are thorough, but they may be somewhat distracting. First, the main experimental results presented in Tables 1 and 2 involve some inconsistent demonstrations; for example, both tables include the "Open-Source LLM-as-a-Judge" part, but the types of methods involved seem different. In Table 1, it' s unclear whether "Ref-Based" refers to references generated by the corresponding LLMs or the original references, which is important. And Sections 4.3 and 4.4 may not be as critical and could be moved to the appendix, given the availability of stronger evaluation methods; this would allow space for more in-depth experiments and analysis.”**
>
> Thanks for carefully pointing out this issue. We completely agree with your advice. We will modify the layout following your advice. In summary,
>
> -        Table 1 shows NLG-Evaluation tasks, where these tasks all include the human-labeled references, "Ref-Based" refers to the original references;
> -        Table 2 shows Instruciton-Following Preference Tasks, where these tasks don't have references, "Ref-Based" refers to the references generated by the LLMs (to ablation study the effectiveness of RevisEval).
> -        For Sec 4.3 and 4.4, we directly verify how the effectiveness of references generated by RevisEval. We agree with your suggestions, and we will reduce the space of Sec.4.3 and Sec.4.4; for example, we will convert the figure to a table.
>
>
> ##### **References**
>
> [1]. Not All Metrics Are Guilty: Improving NLG benchmarks by Diversifying References, In NAACL' 24
>
> ---
>
> We hope our response helps the reviewer better understand how we think about this work. Any future discussion and comments are more than welcome.

---

> ### Author Response · Authors · 2024-11-23
> **Appreciate any feedback**
>
> Dear Reviewer 6W2K,
> As the discussion phase is coming to a close soon, we look forward to hearing from you and would greatly appreciate any feedback you can provide. Your insights would be invaluable in helping us improve the quality of our paper. Thanks!

---

> > ### Comment · Reviewer_6W2K · 2024-11-24
> >
> > Thank you for your detailed response, as well as the corresponding clarifications and additional experiments. I’m glad that some suggestions have been implemented and have brought practical improvements. Considering the significant extent of the required modifications, I have made an appropriate adjustment to the score.

---

> > > ### Author Response · Authors · 2024-11-24
> > >
> > > Dear Reviewer 6W2K,
> > > We are deeply appreciative of your time and effort! Your feedback has been incredibly valuable in enhancing our paper, and we are sincerely thankful for your insights.  If you still have any other questions in the future, please feel free to let us know. We will continue to try our best to answer for you.

---

### Official Review · Reviewer_9THu · 2024-11-04

**Soundness:** 4
**Presentation:** 4
**Contribution:** 3
**Rating:** 8
**Confidence:** 4

**Summary:**

Recently, LLM-as-a-Judge has been gaining popularity for evaluating language generation tasks, but still has a few reliability challenges compared to human evaluation. This paper proposes RevisEval, a text evaluation method that can be used for LLM-as-a-Judge methods as well as more traditional reference-based evaluation metrics, such as BLEU and BERTScore. The core of the method is to use LLMs to revise the response (system output) based on the human reference, called reponse-adapted references, which is then used as a new reference in the downstream evaluation, be it LLM-as-a-Judge or traditional evaluation metrics. Through experiments, the authors showed that 1) the proposed method RevisEval showed improved correlation with gold standard compared to reference-free and baseline reference-based evaluation methods, and 2) on preference tasks, RevisEval outperforms baselines including fine-tuned LLM-as-a-Judge models, 3) the proposed method reduces the positional bias compared to reference-free methods as well as conventional reference-based method.

**Strengths:**

* Simple yet effective method — the core idea of the proposed method, RevisEval, is very simple—simply "rewrite" the response based on the human-written reference (and the rubric) and use it as a new reference. The method is also effective for many settings including LLM-as-a-Judge and traditional reference-based metrics. It is easy to imagine that the proposed method is used in evaluation of many NLG tasks going forward.
* Good ablation studies — the paper provides a wide set of ablation studies to show the proposed method's effectiveness. It shows evaluation results on scoring tasks as well as pairwise preference benchmarks. I also liked the bias analysis (Section 4.5) as well as the detailed analysis of concrete examples (Section 5).

Overall, the paper is overall well written and provides enough evidence that the proposed method is simple, effective, and widely applicable.

**Weaknesses:**

No major weakness as far as I see. Here are some minor weakness points:

* Unclear names—personally I find "response-adapted references" very confusing. It sounds like the method adapt references based on response, but actually it's the other way around. It is actually reference-adapted responses, but I'm not sure if this is a better way of describing it (I don't have any better ideas).

* Unclear description of the experiment settings—the main body of paper benefits a bit of description about the benchmarks. It is based on Tigerscore, but the paper provides very little information re: the specific datasets used and their sizes. Importantly, I think the variety and the quality distribution of responses matter a lot for the evaluation of evaluation methods, and a few sentences about the quantity and the quality of the benchmark datasets would be very helpful.

* Future prospect—this is a bit hypothetical, but the very reason why RevisEval works at all is that current LLMs are in general better at generation rather than discrimination, as the authors state in Section 4.4. Does this mean that in the future, if we have more powerful LLMs at discrimination, would the proposed method still be useful, since future LLMs can simply "guess better" using the reference and the response?

* Applicability — the paper already shows experimental results on a wide range of tasks and benchmarks, but I'm suspecting they are all English tasks (only exception is the source sentences of MT, which are in Chinese). It doesn't have to be done in this paper, but it would be valuable to test the effectiveness of RevisEval in a wider range of tasks (e.g., image captioning) and languages.

**Questions:**

* How would the proposed method work for multiple references? For open-ended text generation tasks, including MT, multiple references are often used.
* What's the metric used in Table 2? Accuracy?
* Figure 3 — it looks like which metrics are most effective (and closest to GPT-4 performance) vary based on the specific metrics used. Would you provide some general guidelines which metric(s) are most effective in general when combined with RevisEval? Or simply doing majority voting is a good strategy?
* In the last paragraph of Section 4.2 — "significantly" is used two times. Are they used in a statistical sense? If not, they are simply very subjective adverb and I would advise against using it in this context
* Same for the section title of 4.3 — what do you exactly mean by "activating?" I would rephrase with something simpler, e.g, "Improving"
* What are some examples of future work of RevisEval? The conclusion section only provides the summary of the findings.

---

> ### Author Response · Authors · 2024-11-17
> **The Response to Reviewer 9THu (1/2)**
>
> Hi Reviewer #9THu, we appreciate your positive and important review about our work. Below is our specific response to your concern.
>
> ---
>
> > #####   **W1: “Unclear names.personally I find "response-adapted references" very confusing. …  describing it (I don't have any better ideas). ”**
>
> Thank you for pointing out the issue of unclear naming. Specifically, we revise the response and then get a (post-generated) reference. Since the revision directly modifies the response, the reference becomes naturally adapted to the response. We will follow your suggestion to reduce any potential confusion and propose a clearer alternative name.
>
> > #####  **W2: “Unclear description of the experiment settings—the main body of paper benefits a … would be very helpful.”**
>
> Thanks for your valuable suggestions about the unclear description of the experiment setting. Actually, we have included these descriptions in the appendix, e.g., Appendix D,E,F. However, we fully agree with your suggestion that the main body requires clearer descriptions. In the revised version, we will modify the layout by incorporating essential descriptions directly into the main text and making it self-contained.
>
> > #####  **W3: “Future prospect—this is a bit hypothetical, but the very reason why RevisEval works at all is that current ... since future LLMs can simply "guess better" using the reference and the response?”**
>
> Thanks to the reviewer for bringing up future prospects and related concerns, and this aspect is worth discussing. Current LLMs are trained and generate text based on the *next-token prediction paradigm*. If continues to develop within this paradigm, we expect that the generation ability will remain central in the LLM. Therefore, the issues you mentioned may not arise unless this LLM paradigm is completely overturned.
>
> Moreover, our method does not conflict with the future prospect of LLMs' discriminative abilities becoming stronger. RevisEval leverages generation capabilities to aid in discrimination effectively. It provides better references for following discrimination, if LLM's discriminative ability is strongly powerful, it can fully utilize the references generated by RevisEval.
>
> Therefore, we authors have confidence in RevisEval's effectiveness on future LLMs.
>
>
> > ##### **W4:” Applicability — the paper already shows experimental results on a wide range of tasks and benchmarks, ... . It doesn't have to be done in this paper, but it would be valuable to test the effectiveness of RevisEval in a wider range of tasks (e.g., image captioning) and languages.“**
>
> Thank you for this suggestion. You provide us an intriguing and exciting insight, and we can envision the potential of the scenario you mentioned. It may indeed be possible to use MLLM to 'revise' multimodal output. This approach could inspire numerous evaluation domains.
>
>
> > ##### **Q1: How would the proposed method work for multiple references? For open-ended text generation tasks, including MT, multiple references are often used.**
>
> Thank you very much for this question, which can help us further expand the applicability of our method. We give the related experiment in general response. As we reported in GR1, multiple references can help our method further improve its effectiveness. So, our proposed method works for multiple references; we will update it in the future version.
>
> > ##### **Q2: What's the metric used in Table 2? Accuracy?**
>
> Yes, the metric is accuracy. The benchmarks in Table 2 are all preference prediction tasks, where the llm-as-a-judge to choose the better one. So, the metric is accuracy. We will clear the description in the updated version.
>
> > ##### **Q3: Figure 3 — it looks like which metrics are most effective (and closest to GPT-4 performance) vary based on the specific metrics used. Would you provide some general guidelines which metric(s) are most effective in general when combined with RevisEval? Or simply doing majority voting is a good strategy?**
>
> Thanks for your great question about the further conclusion about which metric(s) are most effective. Our proposed guidelines are:
>
> i) No single metric has a consistent superiority in all benchmarks, and Moverscore looks like a relatively best metric among all metrics;
>
> ii)  We advocate majority voting, because it has a stably and consistently superior performance.
>
> We will update this clarification in the next version following your advice.
>
> > ##### **Q4: In the last paragraph of Section 4.2 — "significantly" is used two times. Are they used in a statistical sense? If not, they are simply very subjective adverb and I would advise against using it in this context**
>
> Thanks for your suggestion and we totally agree with your advice. They are subjective adverbs. We will modify the wording in the updated version.

---

> ### Author Response · Authors · 2024-11-17
> **The Response to Reviewer 9THu (2/2)**
>
> > ##### **Q5: Same for the section title of 4.3 — what do you exactly mean by "activating?" I would rephrase with something simpler, e.g, "Improving"**
>
> We appreciate your suggestion, and we agree that ‘Improving'  might be more simpler. We will change accordingly in future versions.
>
> > ##### **Q6: What are some examples of future work of RevisEval? The conclusion section only provides the summary of the findings.**
>
> Thank you for your valuable advice. We are confident in the future examples of our work:
>
> i) **New Paradigm**: Reviser + Classic Metric – a completely new evaluation paradigm that can benefit the community, particularly for weak or small LLMs.
>
> ii) **New Domain**: Multi-modal – As you kindly suggested in W4, the revision mechanism is not limited to LLMs; it can also be applied to image generation models, such as similar mechanisms like denoising.
>
> iii) **New Pipeline**: Multi-agents – We have demonstrated that the reviser is a useful agent that can be integrated into the evaluation pipeline. In future powerful multi-agent setups, our proposed RevisEval can be incorporated as an essential component.
>
> We will supplement this in the conclusion section in the updated version.
>
> ---
>
> We hope our response helps the reviewer understand how we think about this work better, and we welcome the reviewer to communicate with us more about it and help us revise it.

---

> ### Author Response · Authors · 2024-11-23
> **Appreciate any feedback**
>
> Dear Reviewer 9THu,
> As the discussion phase is coming to a close soon, we look forward to hearing from you and would greatly appreciate any feedback you can provide. Your insights would be invaluable in helping us improve the quality of our paper. Thanks!

---

### Official Review · Reviewer_Rcn4 · 2024-11-09

**Soundness:** 1
**Presentation:** 4
**Contribution:** 3
**Rating:** 5
**Confidence:** 5

**Summary:**

The paper proposes an interesting method “RevisEval” which explores a new approach to performing reference-based evaluation by modifying references based on the responses to be evaluated. The authors show that this improves the reliability of LLM-based evaluators as compared to using static references, by hypothesizing that an effective reference must be closely relevant to the response to be evaluated. Authors show many interesting observations and analysis across various standard NLG tasks as well as open-ended generative tasks and also evaluate various metrics (both standard and LLM-based). Authors also show that these adapted references can even boost the efficacy of standard metrics.

**Strengths:**

1. The paper motivates the problem very well by identifying the issues with current reference-based evaluation paradigms. The idea of dynamically generating contextually relevant references is creative and interesting. It aims to address very important and quite relevant aspects of using LLM as evaluators.
2. Extensive experiments have been conducted across different tasks as well as various metrics have been evaluated. The authors also show the generalizability of their approach to different metrics.
3. The paper also considers and accounts for the various biases present in LLM Evaluators and also considers the cost of conducting evaluations (which is often ignored in a lot of works)
4. Many interesting insights have been reported by the authors, including using these contextually relevant references to improve the standard n-gram and model-based metrics.

**Weaknesses:**

While I agree with the motivation behind the paper, I am not sure about the soundness of the  methodology followed to generate the reference answers:
1. Using the response itself to generate an "adapted reference", the evaluation might indirectly validate the response’s content and structure. This may lead to artificially inflated evaluations, as the evaluator is essentially comparing the response against a modified version of itself, which serves as the reference.
2. If the response contains subtle errors, the adapted reference “might” effectively validate or normalize these errors. These is no study around whether the reviser indeed accounts for or corrects for these errors.
3. While this approach may work well for evaluations of standard NLG tasks as well as some open-ended tasks that care about the language generation capabilities, but for evaluations that care about the factual accuracy of the responses (something where LLMs are overall known to hallucinate), this simple revision may not be robust.

**Questions:**

1. While the overall paper is well-written, mentioning what the numbers mean in each table and how they have been calculated in the captions or in the text may improve the readability of the paper to a general user. For eg:  mentioning that the values in Table 2 is the accuracy against human preferences...
2. As mentioned in the weaknesses, please provide details of any experiments that were conducted to study the soundess of this approach for factual responses (where the generated response contains errors which get normalised in the adapted reference).

---

> ### Author Response · Authors · 2024-11-17
> **The Response to Reviewer Rcn4 (1/2)**
>
> Hi Reviewer Rcn4, we thank you for the constructive review. Below is our specific response to your concern.
>
> ---
>
> > #####   **W1: “Using the response itself to generate an "adapted reference", the evaluation might indirectly validate the response' s content and structure. This may lead to artificially inflated evaluations, …  the reference.”**
>
> Thank you for your review and concern regarding ''artificially inflated evaluation.'' Based on our humble understanding, this ''inflated evaluation'' means that since the reference is derived from revising the response, the evaluation results (absolute score rating) might be biased (inflated high) toward the response. If our understanding is correct, RevisEval remains unaffected by this issue:
>
> i) In **pairwise-comparison** judge, where the goal is to pick the better one between two responses, RevisEval randomly selects one of the responses to be revised as the adapted reference. This strategy avoids systematic bias since there is no consistent advantage for either response. So ''artificially inflated evaluation'' will not undermine the soundness of our evaluation system.
>
> ii) In **score rating**, the goal of automated evaluation using reference is to match with human evaluation ratings. We use correlation (e.g., Pearson, Spearman correlations) to measure this matching. If 'Artificially inflated evaluation' is significant, the correlation will be lower. Therefore, 'Artificially inflated evaluation' does not affect the soundness of any metric at the system level, including RevisEval.
>
> we hope our clarification can give you more confidence in the soundness of RevisEval.
>
> > #####  **W2: If the response contains subtle errors, the adapted reference “might” effectively validate or normalize these errors. These is no study around whether the reviser indeed accounts for or corrects for these errors.”**
>
> We appreciate the reviewer for mentioning the important aspect of error. As the reviewer stated, the revision process may fix these errors in the response-adapted reference. In our RevisEval, the response-adapted reference is only used for a better evaluation, trying to align with human evaluators'  preferences. Unlike BLEU or ROUGE, response-adapted reference is not used as the golden standard, and it is a guidance/hint to aid the following evaluation.
>
> In addition, the experiments on LLMBar and using BLEU/ROUGE validate this issue to some degree: (i) LLMBar Benchmark: the judge' s performance on LLMBar depends on whether the judge can detect subtle errors in the instruction following. As reported in Table 2, RevisEval improves the llm-as-a-judge, proving our references effectively validate these errors;  (ii) BLEU/ROUGE: BLEU/ROUGE (similarity calculation) totally relies on references to capture subtle errors. In Sec.4.3 and 4.4, we also input the references generated by RevisEval to the BLEU/ROUGE to verify our references. To compute accuracy and the correlation to human scores, we have demonstrated that our references effectively validate these errors.
>
> More importantly, we fully agree with your suggestion to see more direct study and evidence. To address this, we provide two studies:
>
> i) **Case Study on LLMBar**:
> #### The First Case
> - _Question:_ Convert from celsius to fahrenheit. Temperature in Celsius: 15
> - _Response:_ 77\u00b0F
> - _Response-Adapted Reference:_ 59\u00b0F
> #### The Second Case
> - _Question:_ Sort them in lexicographical order: \n\napples, grapes, bananas, oranges, kiwi, pears, plums, oranges, cherries, apples, tangerines, peaches
> - _Response:_ apples, bananas, cherries, grapes, kiwi, oranges, peaches, pears, plums, tangerines
> - _Response-Adapted Reference:_ apples, apples, bananas, cherries, grapes, kiwi, oranges, oranges, peaches, pears, plums, tangerines
>
> We can observe that compared with the response, the adapted reference contains fewer subtle errors. This might make llm-as-a-judge easier to evaluate, as containing errors directly affects the quality of the response, leading to a higher alignment with human evaluators than reference-free evaluators.
>
> ii) **Direct Quality Evaluation on Response-adapted References**: we directly score the quality (1~5) of the adapted references on **correctness** aspect compared to the original responses using LLM-as-a-Judge.
>
> ||Adversarial_Neighbor|Adversarial_GPTInst|Adversarial_GPTOut|Adversarial_Manual|Natural|Overall|
> |---|---|---|---|---|---|---|
> |Response 1|3.27|3.03|3.47|3.39|2.99|3.19|
> |Response 2|3.16|3.01|3.32|3.47|3.44|3.25|
> |Response-adapted Reference|4.72|4.73|4.53|4.91|4.82|4.75|
>
> On 5 subsets of LLMBar, the references (revised by RevisEval-gpt-4-turbo) have consistently better correctness than the responses, which means RevisEval detects and corrects the subtle errors.
>
> We will supplement this study in the updated version.

---

> ### Author Response · Authors · 2024-11-17
> **The Response to Reviewer Rcn4 (2/2)**
>
> > #####  **W3: While this approach may work well for evaluations of standard NLG tasks as well as some open-ended tasks that care about the language generation capabilities, but for evaluations that care about the factual accuracy of the responses (something where LLMs are overall known to hallucinate), this simple revision may not be robust.**
>
> > #####  **Q2: As mentioned in the weaknesses, please provide details of any experiments that were conducted to study the soundess of this approach for factual responses (where the generated response contains errors which get normalised in the adapted reference).**
>
> Firstly, we propose a method for improving general judge performance, so we did not focus on specific aspects, including factual accuracy. However, we fully agree that your emphasis on factual accuracy is very important, especially as you may have concerns about whether the revision mechanism can correct factual errors. Here, we provide two points of evidence:
>
>  i). **Related Work**:  [1,2] presents that the (post) revision is a highly effective mechanism for correcting factual errors, such as hallucinations;
>
> ii). **Experiment**: We verify RevisEval on JudgeBench [3], specifically focusing on evaluating the Judge' s performance regarding the factual accuracy of the responses.
>
> |Method|knolwedge|math|reasoning|coding|overall|
> |---|---|---|---|---|---|
> |Vanilla Judge(gpt-4-turbo)|48.05|69.64|56.12|38.09|52.57|
> |RevisEval(gpt-4-turbo)|64.29|64.29|70.41|45.24|63.71|
> |---|---|---|---|---|---|
> |Vanilla Judge(gpt-4o)|53.2|55.4|49.0|35.7|50.3|
> |RevisEval(gpt-4o)| 72.7|58.9|66.3|33.3|64.0|
> |---|---|---|---|---|---|
> |Vanilla Judge(gpt-4o-mini)|64.3|62.5|65.3|42.9|61.7|
> |RevisEval(gpt-4o-mini)|70.8|64.3|63.3|54.8|65.7|
>
> RevisEval demonstrates strong performance in evaluating factual correctness across different domains.
>
> We will also update this analysis in the next version and hope you do not worry about this concern.
>
> > #####  **Q1: While the overall paper is well-written, mentioning what the numbers mean in each table and how they have been calculated in the captions or in the text may improve the readability of the paper to a general user. For eg: mentioning that the values in Table 2 is the accuracy against human preferences…**
>
> The reviewers' constructive and detailed suggestions are extremely valuable to us. We will clear and polish the detailed description better.
>
> ##### **References**
> [1] Automatically Correcting Large Language Models: Surveying the Landscape of Diverse Automated Correction Strategies, TACL' 24
>
> [2] RARR: Researching and Revising What Language Models Say, Using Language Models, In ACL' 23
>
> [3] JudgeBench: A Benchmark for Evaluating LLM-based Judges, Arxiv.2410.12784
>
> ---
>
> We hope our response helps the reviewer understand how we think about this work better. We welcome the reviewer to communicate with us more and help us revise the paper.

---

> ### Author Response · Authors · 2024-11-23
> **Appreciate any feedback**
>
> Dear Reviewer RCn4,
> As the discussion phase is coming to a close soon, we look forward to hearing from you and would greatly appreciate any feedback you can provide. Your insights would be invaluable in helping us improve the quality of our paper. Thanks!

---

> ### Author Response · Authors · 2024-11-28
> **Thanks for your response and further discussion! (1/4)**
>
> We sincerely appreciate your detailed feedback, particularly your careful elaboration of your concerns. This has allowed us to better understand your concerns and provide a more precise response to address them.
>
> ### **The general idea of RevisEval**
>
> First, we would like to reiterate the core challenges our method aims to solve, which are closely related to the concerns you raised about *bias* and *inflated scores*.
>
>
> **Why Does Our Method Work?** As illustrated in Figure 1 (in Lines 52-74 of our manuscript), when we use GPT-4-generated answer-as-references, these references will decrease evaluation effectiveness as their relevance to response decreases. This motivates us to seek a mechanism to create relevant references to improve the LLM-as-a-Judge. We observe when LLM generates the response-adapted references, LLMs tend to modify low-quality text segments in the response with high probability while retaining high-quality segments. This behaviour originates from *the inherent training objective of LLMs*—LLMs are predominantly trained to generate high-quality, human-like outputs, with limited exposure to the distribution of low-quality or flawed data. This observation is also utilized by numerous studies[1-4] that post-revision by LLMs is a reliable mechanism to improve response quality. The revision process of LLMs mirrors human revision habits: the greater the extent of modification, the lower the original quality of the text is likely to have been. Leveraging this revision mechanism, the references generated through our method effectively support evaluations.
>
> ### **Detailed Responses to your specific concern**
>
> #### 1) **questions about the independence of the reference.**
>
> We thank the author for bringing up the issue of the **independence between reference and response**. As mentioned above, it proves that **the relevance to the response** is the key factor in determining whether the reference is beneficial for evaluation, which serves as the intuition of RevisEval. We presume that the reviewer believes if written independently, the human/LLM-generated response and reference are independent. If we make the right assumption, in Open-ended tasks, like Story Generation and Instruction-following tasks, human/GPT-4 references are mostly **irrelevant but independent** references with responses, and these references will decrease the LLM-as-a-Judge evaluation performance due to irrelevance. Our novelty is to challenge traditional assumptions about creating references paradigm and find the key factor to enhance the reference's effectiveness.

---

> ### Author Response · Authors · 2024-11-28
> **Thanks for your response and further discussion! (2/4)**
>
> #### 2) **Assumptions about adapted references could inadvertently reflect factual errors in the incorrect response.**
>
> We thank the reviewer for describing a detailed case, which has helped us accurately understand your concern. We sincerely appreciate the reviewer's patience again.
> First, we acknowledge that imperfect revisions may occur, and such cases are possible. However, the key question is whether this is an extreme case or a common phenomenon. To address this, we provide both a case explanation and empirical evidence.
>
> ##### a) **Case Explanation**
> As mentioned above, the revision process in LLMs is not a simple copy operation. Again, We emphasize that our revision mechanism **incorporates information from two responses rather than solely one response being revised** (shown in Lines 177-184 of the manuscript).
>
> Due to the training objective of LLMs—LLMs are predominantly trained to generate high-quality, human-like outputs—the probability of retaining correct segments is significantly higher than that of retaining erroneous segments during the revision process.
>
> For example, let *E* represent erroneous segments in the response, and *H* represent high-quality segments. Consider two responses: $R_1: E_1, H_2, H_3$; $R_2: H_2, H_3, H_4$ and $R_2$ has a higher quality. After revision, the reference $R^\*$ is likely to retain $H_2$, $H_3$, and $H_4$ while eliminating $E_1$. Then LLM-as-a-Judge will pick the $R_2$, as $R^*$ would align more with high-quality segments overall. If the LLM fails to remove $E_1$ and retains it, the resulting $R^\* = E_1, H_2, H_3, H_4$ would be equally closer to $R_1$ and $R_2$. Then, inputting the reference adapted from $R_1$ into LLM-as-a-Judge would make no difference from LLM-as-a-Judge without reference.
>
> We will update this explanation in the revised manuscript.
>
> ##### b）**Empirical Evidence given the confusion matrix**
>
> Here we investigate whether there is a noticeable bias when using either Response 1 or Response 2 as the primary text (detail see in Lines 177-180 of the manuscript) to be revised. Specifically, We aim to examine whether our method leads to significant changes in LLM-as-a-Judge's decisions when different responses are selected as the primary text for generating references. We do so by reporting the confusion matrix[5]. If significant changes are witnessed between the confusion matrix with reference generated based on Response 1 or Response 2, it would indicate that the adapted reference is influenced by choosing which particular response is the primary text for revision, which means the reviewer's concern is a common phenomenon. Otherwise, it suggests this is rather an extreme case. The experiments are performed on three benchmarks: RewardBench, LLMBar, and MTBench.
>
>
> In the confusion matrix, columns represent the predicted responses, while rows represent the ground truth. As we showed in two tables of each benchmark, **the results across both matrices are rather similar, indicating the reviewer's concern is rare in our approach**. Specifically, in RewardBench, when Response 1 is the primary text to be revised, the predictions for Response 1 and Response 2 align well with their respective labelled responses (1357 vs. 1432). Similarly, when Response 2 is the primary text, the alignment remains comparable (1331 vs. 1415). The overall distribution of predicted labels also shows minimal variation across both setups (1361/1524 vs. 1352/1533). These results suggest that the revision process does not introduce a noticeable bias that would cause it to overlook errors systematically, regardless of which response is prioritized.
>
>
> Rewardbench(2194 cases)
>
> Using Response 1 as revision primary text
>
> | ground truth \ predicted        | response 1 | response 2 |
> | ---------- | ---------- | ---------- |
> | response 1 | 1357       | 92         |
> | response 2 | 104        | 1432       |
>
> Using Response 2 as revision primary text
>
> | ground truth \ predicted        | response 1 | response 2 |
> | ---------- | ---------- | ---------- |
> | response 1 | 1331       | 118        |
> | response 2 | 121        | 1415       |
>
>
>
> MTBench(1284 cases)
>
> Using Response 1 as revision primary text
>
> | ground truth \ predicted  | response 1 | response 2 |
> | ------------------------------------------- | ---------- | ---------- |
> | response 1                                  | 547        | 120        |
> | response 2                                  | 105        | 512        |
>
> Using Response 2 as revision primary text
>
> | ground truth \ predicted  | response 1 | response 2 |
> | ------------------------------------------- | ---------- | ---------- |
> | response 1                                  | 553        | 114        |
> | response 2                                  | 92         | 525        |
>
>
> In MT-Bench, the prediction distributions are also close, regardless of which response is chosen as the primary text for revision.

---

> ### Author Response · Authors · 2024-11-28
> **Thanks for your response and further discussion! (3/4)**
>
> *Continued with the above*,
>
> LLMBar(419 cases)
>
> Using Response 1 as revision primary text
>
> | ground truth \ predicted  | response 1 | response 2 |
> | ------------------------------------------- | ---------- | ---------- |
> | response 1                                  | 164        | 45         |
> | response 2                                  | 42         | 168        |
>
> Using Response 1 as revision primary text
>
> | ground truth \ predicted  | response 1 | response 2 |
> | ------------------------------------------- | ---------- | ---------- |
> | response 1                                  | 160        | 49         |
> | response 2                                  | 40         | 170        |
>
>
> The conclusions from the analysis of the other two benchmarks are consistent, confirming that the reviewer's concern, though being a reasonable concern, represents an extreme case rather than a common phenomenon in reality.
>
> Furthermore,
> In the previous response, we introduced JudgeBench, a benchmark specifically designed to evaluate factual errors in responses. It provides empirical evidence that **our evaluation method effectively supports LLM-as-a-Judge, even when the responses contain factual errors**. JudgeBench aligns perfectly with the scenario you described, making it an ideal match for addressing your concerns.
>
> #### 3）**Absolute score inflation**
>
> i) **RevisEval Does Not Cause Absolute Score Inflation**
>
> We would like to clarify that RevisEval does not cause absolute score inflation, and we have provided evidence to support this claim.
>
> Let's re-emphasize our RevisEval; RevisEval first generates a response-adapted reference, then incorporates this reference to the following LLM-as-a-Judge for scoring. Here, LLM-as-a-Judge will **rate a reasonable score by prompting** instead of naively computing the similarity like NLG metrics.
>
> First, we directly compare the absolute scores predicted by RevisEval with those derived from human-labeled evaluations.
> This comparison is a simple validation process, where we observe that our predicted absolute scores are consistently **lower** than the human-labeled scores, rather than inflated.
>
> | |summeval|wmt|data2text|story generation|
> |-|-|-|-|-|
> |human labeled absolute score (mean)|4.13|3.85|4.4|2.51|
> |RevisEval predicted absolute score (mean)|3.89|3.91|3.7|1.83|
>
> Further, we use NLG metrics to validate our approach. Specifically, we compare the absolute scores calculated using human references with those calculated using adapted references, focusing on tasks like translation and summarization. The following tables show the results for BLEU and ROUGE scores:
>
>
> |WMT(translation)|bleu|rouge|
> |-|-|-|
> |human reference|0.2264|0.4918|
> |response-adapted reference|0.2014|0.4476|
> |-|-|-|
> |Summeval(summaraztion)|bleu|rouge|
> |-|-|-|
> |human reference|0.1187|0.3232|
> |response-adapted reference|0.2507|0.4101|
>
> The data clearly shows that **scores based on response-adapted references are not necessarily higher than those based on human references**. In fact, for some tasks, the adapted references yield lower scores than the human references, further demonstrating that RevisEval does not inflate scores.
>
>
> ii) **The Role of Absolute Scores in Evaluation Systems**
>
> It is important to emphasize that **absolute scores are not the metric for evaluating the effectiveness of an evaluation system**. Evaluation systems are not directly comparable based on absolute scores unless they are standardized, as the reference used in the scoring process is an internal factor for each system. For this reason, correlation is the primary metric used to assess evaluation systems, as it captures the consistency and alignment of predictions with human evaluations rather than their absolute values.
>
> For example, consider BLEU scores derived from different reference generation methods: [6] human-generated multiple references and [7] LLM-generated multiple references. The absolute scores produced by these two methods will differ from those derived from a single expert reference, but this difference does not invalidate either approach. Both methods reliably enhance the metric’s performance through the use of references.
>
> Thus, the true value of an evaluation system lies not in the absolute scores it produces but in **its correlation with human-labeled scores across tasks and systems**. If response-adapted references yield higher or lower scores in some cases, this should not be viewed as an indication of an ineffective system.
>
> **Conclusion**
>
> In summary, absolute score inflation is not an issue with RevisEval, and correlation, rather than raw scores, is the true metric for evaluating the performance of an evaluation system.

---

> ### Author Response · Authors · 2024-11-28
> **Thanks for your response and further discussion! (4/4)**
>
> #### 4) **Quality Rating**
>
> We apologize for not clarifying the scoring model earlier. The scoring model we used is GPT-4o, while the revising model is GPT-4-turbo; **they are not the same model**. Therefore, this scenario does not align with the concern that a model might favour its own responses.
>
> However, we admit your concern is highly valuable. Associated with W2 about ''no study around whether the reviser indeed accounts for or corrects for these errors,''
> we introduce a human evaluation experiment with two participants. We randomly select 100 examples from LLMBar and have human evaluators perform blind comparisons between the response and the adapted reference.
> When presented with both Response 1 and the adapted reference, **human evaluators favoured the adapted reference in 82% of cases**. Similarly, when presented with Response 2 and the adapted reference, **86% of human evaluators chose the adapted reference**.
> We can observe that humans tend to favour our references. So the response-adapted references have consistently higher quality.
>
> We are genuinely grateful for your detailed explanation of your concerns, as it allows us to address them more effectively. We hope to earn your confidence in our work, especially as our work seeks to challenge the conventional assumption that references must be independent and predefined. We look forward to receiving your feedback on our response, as it will help us further refine and improve our work.
>
> >### References
> >
> >[1] Re3: Generating longer stories with recursive reprompting and revision, In EMNLP'22
>
> >[2]  RL4F: Generating natural language feedback with reinforcement learning for repairing model outputs, In ACL'23
>
> >[3]  Beyond imitation: Leveraging fine-grained quality signals for alignment, In ICLR'24
>
> >[4] RARR: Researching and Revising What Language Models Say, Using Language Models, In ACL' 23
>
> >[5] Confusion_matrix:   /wiki/Confusion_matrix
>
> >[6] BLEU might be guilty but references are not innocent, In EMNLP'20
>
> >[7] Not All Metrics Are Guilty: Improving NLG Evaluation by Diversifying References, In NAACL'24

---

> > ### Author Response · Authors · 2024-12-04
> >
> > We sincerely appreciate your previous feedback and look forward to your new insights, which have greatly enhanced the quality and clarity of our work. As we finalize our revisions, we remain eager and hopeful for your recognition of our rearticulated contributions. Our work represents a fresh rethinking of the effectiveness of references in evaluation, and we hope it inspires further discussion and exploration in this area, guided by your valuable suggestions.

---

### Author Response · Authors · 2024-11-17
**General Responses**

First, we sincerely thank all reviewers for their constructive comments on improving this paper. Below, we listed some general responses targeting some shared questions.

> ####  **GR1: Whether work for multiple references or more refined-grained references?**

**Q1 of #9Thu**: *''How would the proposed method work for multiple references?'',*

**W2 of #6W2K**: *''I suggest exploring the use of more refined response-adapted references''.*

Thanks to reviewers #9Thu and #6W2K for reminding us about the effectiveness of the multiple/fined-grained references, which can help us further expand the applicability of our method.

We give a general experiment to demonstrate it. In the original RevisEval setting, we revise the response once to generate one reference, wherein the reviser prompt, ''Your revision should consider factors such as the helpfulness, relevance, accuracy, depth, creativity, and level of detail of their responses,'', we include all aspects in one prompt. Now, we conduct separate fine-grained revisions to generate multiple references, where each reviser prompt only includes one aspect (helpful, accuracy, relevance, depth, creativity), then use them in the following evaluation separately. For each case, we will evaluate it based on each reference, and we will get multiple predicted preferences or scores. For preference evaluation task, we will run a majority-voting to get a final preference; for score-rating evaluation task, we will obtain a mean score.  We name this new evaluation pipeline as **Fine-grained RevisEval**, and test its accuracy in the MTBench (preference task) as follows:

||gpt-4-turbo|gpt-4o-mini|
|--|--|---|
|LLM-as-a-Judge|81.18|80.29|
|RevisEval|83.01|81.38|
|Finegrained-RevisEval|84.13|81.99|

So, our proposed method works for multiple/fine-grained references; we will update it in the future version.

>#### **GR2: Further Experiment on RewardBench**

Thanks for the Reviewer #zE3r' s reminder.

Firstly, we want to clarify our method is to improve the LLM-as-a-Judge rather than the Reward Models.

In the W3 of #zE3r, the reviewer has a concern about whether RevisEval works for challenging/difficult question-response pairs. While we have tested on LLMBar, a challenging enough benchmark, to verify this issue, RewardBench[1] is the latest popular challenging benchmark covering more challenging domains, such as CHAT-HARD, REASONING, and SAFETY subsets. So, we decide to choose the rewardbench as an extensive experiment, the result as below:

||CHAT|CHAT-Hard|Safety|Reasoning|Overall|
|-|-|-|-|-|-|
LLM-as-a-Judge(gpt-4-turbo)|97.76|80.04|88.51|90.01|89.04|
Reviseval(gpt-4-turbo)|97.21|81.14|90.01|91.89|89.51|
|-|-|-|-|-|-|
LLM-as-a-Judge(gpt-4o-mini)|96.37|60.09| 91.89|81.21|82.45|
RevisEval(gpt-4o-mini)|93.30|65.13|93.08|86.35|85.63|
|-|-|-|-|-|-|
LLM-as-a-Judge(gpt-4o)|98.60|79.17| 92.03|94.47|91.96|
RevisEval(gpt-4o)|97.76|83.55|93.51|95.53|93.47|

RevisEval can improve LLM-as-a-Judge consistently on RewardBench. Especially in challenging subsets, RevisEval surpasses LLM-as-a-Judge stably.

We promise to update this experiment on our next version.

#### **References**

[1] RewardBench: Evaluating Reward Models for Language Modeling. Arxiv.2403.13787, Citation~99

---

> ### Author Response · Authors · 2024-11-25
> **General Response about Summary of the Updated Manuscript**
>
> We would like to express our sincere gratitude to the reviewers for their valuable feedback, which has greatly helped us improve our paper.
>
> Below, we summarize our revisions to facilitate the reviewers' review process:
>
> > ### **Sec 4.3**: We revised the section from "Activating Classic Metrics Performance" to "Investigation of Response-Adapted References," as per the comments from #Rcn4 W2, W3, Q2, and #9THu Q5.
>
> This change better highlights our aim of investigating the effectiveness and soundness of response-adapted references. While retaining the original experiments, we resized Figure 3 and introduced a direct rating for response-adapted references, comparing their scores with those of the original responses to validate that our method can correct errors and improve quality.
>
> > ### **Appendix J**: The correctness study of response-adapted response
>
> This section supplements Sec 4.3 with case studies and more detailed ratings, aligning with the suggestions from #Rcn4 W2.
>
> > ### **Appendix K**: Evaluation focusing on factual accuracy
>
> To address #Rcn4 W3 and Q2's concerns regarding factual errors in responses, we conducted additional experiments mentioned in Sec 4.3 and detailed in Appendix K.
>
> > ### **Table 2 Caption**:
>
> We revised the caption from "Results of LLM-as-a-Judge on instruction-following preference tasks." to "Accuracy of LLM-as-a-Judge on instruction-following preference tasks." based on feedback from #Rcn4 Q1 and #9THu Q2.
>
> > ### **Sec 4.1 Evaluation Setting**:
>
> Following #9THu W2 and #6W2K W3, we explicitly supplement the corresponding benchmarks in the main text and clarify the inherent differences between reference-based settings in the two tasks.
>
> > ### **Sec 6 Conclusion**
>
> In accordance with #9THu W4 and Q6, we supplemented the applicability discussion with examples of future work.
>
> > ### **Appendix L**: Multiple refined-grained references
>
> To explore the feasibility of multiple references, we added this section as per #9THu Q1 and #6W2K W2.
>
> > ### **Sec 4.4** guideline
>
> We incorporated additional content to address #9THu Q3's suggestions regarding "a guideline".
>
> > ### **Subjective Adverbs**:
>
> We revised subjective adverbs like "significantly" to another adverbs following #9THu Q4's feedback.
>
> > ### **Appendix M**: comparison with div-ref
>
> To compare our work with Div-Ref, we added this section as suggested by #6W2K Q1.
>
> > ### **Appendix N**: performance on rewardbench
>
> To address #zE3r W3's concerns about whether RevisEval works for challenging/difficult question-response pairs, we included experiments on RewardBench.
>
> We have addressed all reviewer concerns and made the best possible revisions as promised in our responses. All changes have been marked in blue in the updated manuscript.
>
> As of now, we are grateful that #6W2K and #zE3r have accepted our responses and revised their scores accordingly.
>
> However, we have not yet received feedback from #Rcn4 and #9THu regarding our responses. Since the public discussion phase will end on November 26th, we are keen to know whether our responses have adequately addressed their concerns. We remain fully committed to providing any further clarifications or updates if needed and look forward to hearing their valuable insights.
>
> Additionally, during #zE3r's discussion process, #zE3r provides a more specific experimental requirement, prompting us to conduct another round of responses and supplementary experiments. As we have not received feedback from #zE3r, we are uncertain whether to include these new experiments in the appendix. We look forward to #zE3r's updated feedback, which will help us further enhance the quality of our paper.

---

> > ### Author Response · Authors · 2024-12-02
> > **General Response about Summary of the Updated Manuscrip (Final)**
> >
> > After receiving clearer explanations from #Rcn4 and #zE3r regarding their concerns, the remaining problems are: "whether adapted references could inadvertently reflect factual errors in the incorrect response," "whether the absolute score is inflated," and "effectiveness on GPQA and Omni." We have provided clear responses to these issues in the new response. Although we have not yet received new feedback from the reviewers, we have fulfilled our commitment by incorporating corresponding updates into the revised manuscript, which have been highlighted in blue. Of course, we still look forward to receiving the reviewers' comments in the final moments of the discussion.
> >
> > > ### **Figure 2**
> >
> > We have added wavy underlines and corresponding explanations in the caption to more clearly illustrate the process of our revision, which retains high-quality segments while revising low-quality parts.
> >
> > > ### **Appendix O: Performance on Challenging Reasoning Benchmarks**
> >
> > We have included our experimental results on GPQA and Omni in the manuscript.
> >
> > > ### **Appendix P: Justification of Using Revision as a Reliable Method to Enhance Text Quality**
> >
> > We have incorporated our justification for the mechanism of revision, as detailed in our responses, into the manuscript.
> >
> >
> > **As the rebuttal period draws to a close, we are deeply grateful for the invaluable feedback that has improved our paper during these discussions. The refinement of this work would not have been possible without the thoughtful suggestions from each reviewer. This journey has been both enriching and enjoyable, and we hope that this will result in a solid and meaningful work.**

---

### Meta-Review · Area_Chair_rCUX · 2024-12-20

**Metareview:**

The paper proposes RevisEval which improves on the standard LLM-as-a-judge pipeline by modifying the reference response to align better with the generated output. Reviewers generally believe the idea has promise and it improves over the baseline LLM-as-a-judge methods. They do raise some issues: (1) revising the LLM output to generate the reference might introduce unintended biases, (2) the improvement over the baseline simply uses a strong LLM to generate a output-agnostic reference is quite minimal. Adding more analysis to study point 1 would strengthen the paper.

**Additional Comments On Reviewer Discussion:**

The main points raised by reviewers are about the unintended biases that might be introduced by using a reference conditioned on the very response it will be used to evaluate. The improvements over the closes baseline (using a LLM generated reference not conditioned on the response) are marginal.

---

### Decision · Program_Chairs · 2025-01-22

Accept (Poster)